# Ba$^{+2}$ ion trapping using organic submonolayer for ultra-low background neutrinoless double beta detector

P. Herrero-Gómez[1,2], J. P. Calupitan ![ORCID][1], M. Ilyn[1], A. Berdonces-Layunta[1,2], T. Wang ![ORCID][1,2], D. G. de Oteyza[1,2,3], M. Corso ![ORCID][1,2], R. González-Moreno[1,2], I. Rivilla ![ORCID][2,3,4], B. Aparicio[4], A. I. Aranburu[5], Z. Freixa[3,5], F. Monrabal[2,3], F. P. Cossío ![ORCID][2,4], J. J. Gómez-Cadenas ![ORCID][2,3], C. Rogero ![ORCID][1,2] ✉ & NEXT collaboration*

If neutrinos are their own antiparticles the otherwise-forbidden nuclear reaction known as neutrinoless double beta decay can occur. The very long lifetime expected for these exceptional events makes its detection a daunting task. In order to conduct an almost background-free experiment, the NEXT collaboration is investigating novel synthetic molecular sensors that may capture the Ba dication produced in the decay of certain Xe isotopes in a high-pressure gas experiment. The use of such molecular detectors immobilized on surfaces must be explored in the ultra-dry environment of a xenon gas chamber. Here, using a combination of highly sensitive surface science techniques in ultra-high vacuum, we demonstrate the possibility of employing the so-called Fluorescent Bicolor Indicator as the molecular component of the sensor. We unravel the ion capture process for these molecular indicators immobilized on a surface and explain the origin of the emission fluorescence shift associated to the ion trapping.

The rare neutrinoless double beta ($\beta\beta 0\nu$) decay, $(Z,A) \rightarrow (Z+2,A) + 2e^-$, can occur if and only if neutrinos are Majorana particles[1], i.e., identical to their antiparticles. An unambiguous observation of such a decay would have deep implications in particle physics and cosmology[2–6]. The conventional double beta decay ($\beta\beta 2\nu$), in which two neutrinos are emitted in addition to the electrons, occurs in a handful of isotopes, some of which also offer the necessary features (a reasonable isotopic abundance, a decay energy sufficiently high, etc.) to be used as sources/targets in experiments seeking to observe the $\beta\beta 0\nu$ decay. Examples of such isotopes which have been used for large-scale searches are $^{76}$Ge, $^{130}$Te, and $^{136}$Xe. All attempts to detect a signal so far have not succeeded, and experimental bounds in excess of $10^{26}$ years have been set for the most sensitive searches based on $^{136}$Xe and $^{76}$Ge[7–10].

The field is currently aiming to increase the sensitivity by at least one, and eventually two or more orders of magnitude[11]. This, in turn, implies large exposures, measured in ton-years, and even more importantly, a greatly enhanced capability to suppress radioactive backgrounds.

In a high-pressure xenon gas time projection chamber (HPXe-TPC) such as those being developed by the NEXT experiment[12,13], the double beta decay of $^{136}$Xe will create a $^{136}$Ba$^{+2}$ dication and two electrons, $^{136}$Xe $\rightarrow$ $^{136}$Ba$^{+2} + 2e^- + (2\nu)$. The decay mode with two neutrinos, $\beta\beta 2\nu$, has been observed in xenon with a lifetime of the order of $2 \times 10^{21}$ years[14]. The signal is the same as that of $\beta\beta 0\nu$, except for the total energy of the electrons, which is a continuous distribution for the $\beta\beta 2\nu$ case and spikes around the decay energy, $Q_{\beta\beta}$ (about 2.45 MeV in the case of $^{136}$Xe) for $\beta\beta 0\nu$[15]. The excellent energy resolution of NEXT

[1]Centro de Física de Materiales (CSIC-UPV/EHU), San Sebastián E-20018, Spain. [2]Donostia International Physics Center (DIPC), San Sebastián E-20018, Spain. [3]Ikerbasque, Basque Foundation for Science, Bilbao E-48009, Spain. [4]Department of Organic Chemistry I, Centro de Innovación en Química Avanzada (ORFEO-CINQA), University of the Basque Country (UPV/EHU), San Sebastián E-20018, Spain. [5]Department of Applied Chemistry, University of the Basque Country (UPV/EHU), San Sebastián E-20018, Spain. *A list of authors and their affiliations appears at the end of the paper. ✉e-mail: celia.rogero@csic.es

allows suppressing the contamination of $\beta \beta 2\nu$ to $\beta \beta 0\nu$ by at least nine orders of magnitude[15-17], and thus, $\beta \beta 2\nu$ is not a significant background for NEXT up to lifetimes of $\beta \beta 0\nu$ in the order of $10^{30}$ years.

In the HPXe-TPC, the electric field that drifts the ionization electrons from the two emitted electrons towards the detector anode, will cause the Ba$^{+2}$ dication to drift towards the cathode[18]. Thus, it is feasible to trigger on interesting events (e.g., those with sufficiently high energy). The barycenter of the electron tracks can be used to predict the impact position of the ion, or alternatively, ion transport devices, known as RF carpets could be used to direct the ion to a specific region in the cathode[19]. Furthermore, it is possible to correlate the arrival time of the dication to the cathode with the electrons detected in the anode. Such coincidence could lead to a virtually background-free experiment.

Although the Barium tagging concept was proposed more than three decades ago[20], a practical way to detect Ba$^{+2}$ in situ in a HPXe-TPC was only conceived recently[21,22]. The idea relies on the capability of fluorescent molecules of changing their optical properties upon detecting target analytes[23,24]. An initial proof of concept[25] resolved individual Ba$^{+2}$ dications in aqueous solution using Fluo-3, a well-known commercial indicator. A suitable Barium detector in NEXT requires a functionalized surface that must include a monolayer of the molecular sensor and must efficiently operate in a noble gas atmosphere. Over the last three years an intense R&D program to develop chemosensors able to form a supramolecular complex with Ba$^{+2}$ in dry medium has been carried out[15,26,27]. To date, no experiments have been conducted in which the processes of chelation and detection occur fully under such conditions.

One of the sensors developed by NEXT, the so-called Fluorescent Bicolor Indicator (FBI)[15], combines an enhanced fluorescence with a shift of the emission spectrum (about 61 nm towards the blue) when the indicator is complexed with Ba$^{+2}$. This is due to the specific molecular design of the fluorescent indicator having a crown ether connected to a benzo[*a*]imidazo[5,1,2-*cd*]indolizine fluorophore by a phenyl group. The benzoimidazoindolizine group has been shown to have highly tunable bright emission[28,29] while the crown ether is capable of interacting with Ba$^{+2}$ ions[23,30]. In the presence of a Ba$^{+2}$ ion, calculations predict that coordination happens between the cation, one nitrogen atom of the fluorophore, the crown ether and the para-substituted phenyl ring, causing a torsion-induced decoupling between non-coplanar components of the fluorophore with respect to the non-chelated molecule. This induces a large change in the electronic properties of the dye, causing a blue-shift in emission, which can be used to filter the signal of chelated indicators.

All the indicators developed by the NEXT experiment are based on crown ethers. Because of their capability to capture a variety of guest species, including metal cations, protonated species, and neutral and ionic molecules[31], crown ethers[32] have been extensively used to recognize and trap metal or molecular ions[30,33]. However, they have been poorly studied in solid state. Few examples can be found in the literature where self-assembled monolayers of crown ether derivatives have been grown and used on surfaces. Moreover, in all previous studies, either the growth or ion trapping or both have taken place in solution[34-36]. There are only two works, as far as we know, where crown ethers were deposited under UHV conditions[37] and their metal trapping capability was proven also under UHV[38]. Thus, in addition to the progress relevant for a future $\beta \beta 0\nu$ experiment, the work presented here advances substantially the understanding of the physico-chemical properties of crown ethers immobilized on solid surfaces.

In this paper we combine two highly sensitive surface techniques: X-ray Photoemission Spectroscopy (XPS) and Scanning Tunnelling Microscopy and Spectroscopy (STM/STS), to prove how different ions interact with FBI molecules deposited on suitable substrates. We demonstrate that Ba$^{+2}$ ions induce molecular conformation changes, modifying the electronic structure that would affect the fluorescence emission at suitable surfaces. Coordination with crown ether happens entirely in ultra-high vacuum (UHV) (see model in Fig. 1), which ensures that chemical, structural and electronic changes happen in the absence of solvents, air molecules or spurious contaminants. This is, therefore, a crucial step toward the development of a Ba$^{+2}$ detector.

## Results and discussion
### Molecular sublimation in vacuum
In order to characterize the ion trapping capability of the FBI molecules, first it is mandatory to deposit these FBI molecules on a surface by sublimation in UHV and characterize them structurally and chemically. Thus, we chose a gold substrate, the Au(111) face (single crystal cut to present (111) close-packed planes parallel to the surface), because it is a well-known noble metal surface where very low substrate-molecule interaction was expected. We used XPS to determine their chemical composition. Since XPS sensitivity is limited to a depth of a few nanometers, molecular coverages below 1 monolayer (ML) were always used in this study. This ensures access to the substrate core levels for calibration.

Figure 2a shows the XPS spectra of the three molecular core levels of FBI, i.e., O 1*s*, N 1*s*, and C 1*s*. The spectra were measured for 0.6 monolayer (ML) of FBI deposited on Au(111) at room temperature (RT). The C 1*s* core level can be fitted using two components, one centered at around 284.7 eV and a second and more intense one at 286.3 eV. The former component corresponds to C−C bonds whereas the component at higher binding energy (BE) includes contributions from C−O and C−N bonds, with their relative intensities in agreement

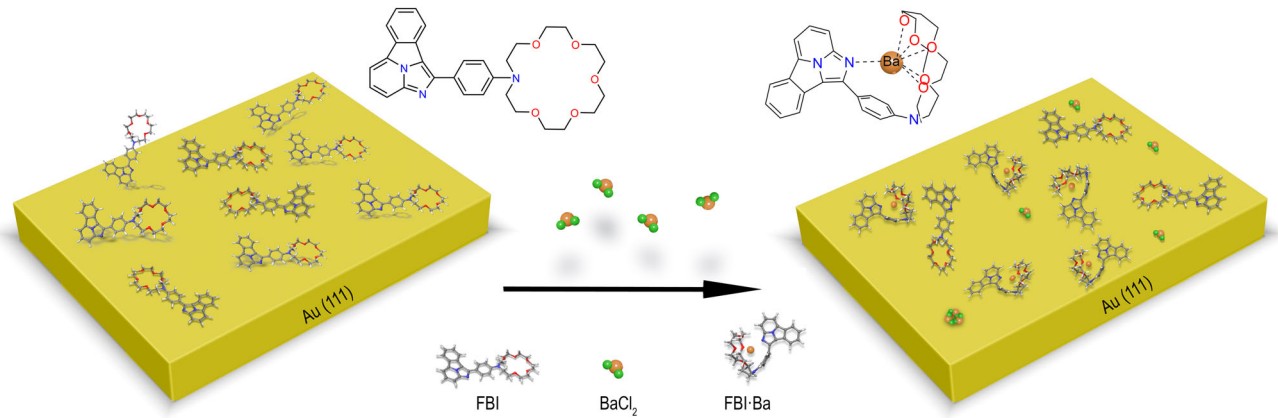

**Fig. 1 | Model of the FBI molecules before and after chelation.** Schematic representation of the experiment we have carried out. FBI molecules were sublimated on a Au(111) surface, chelated in situ and characterized inside the UHV chamber.

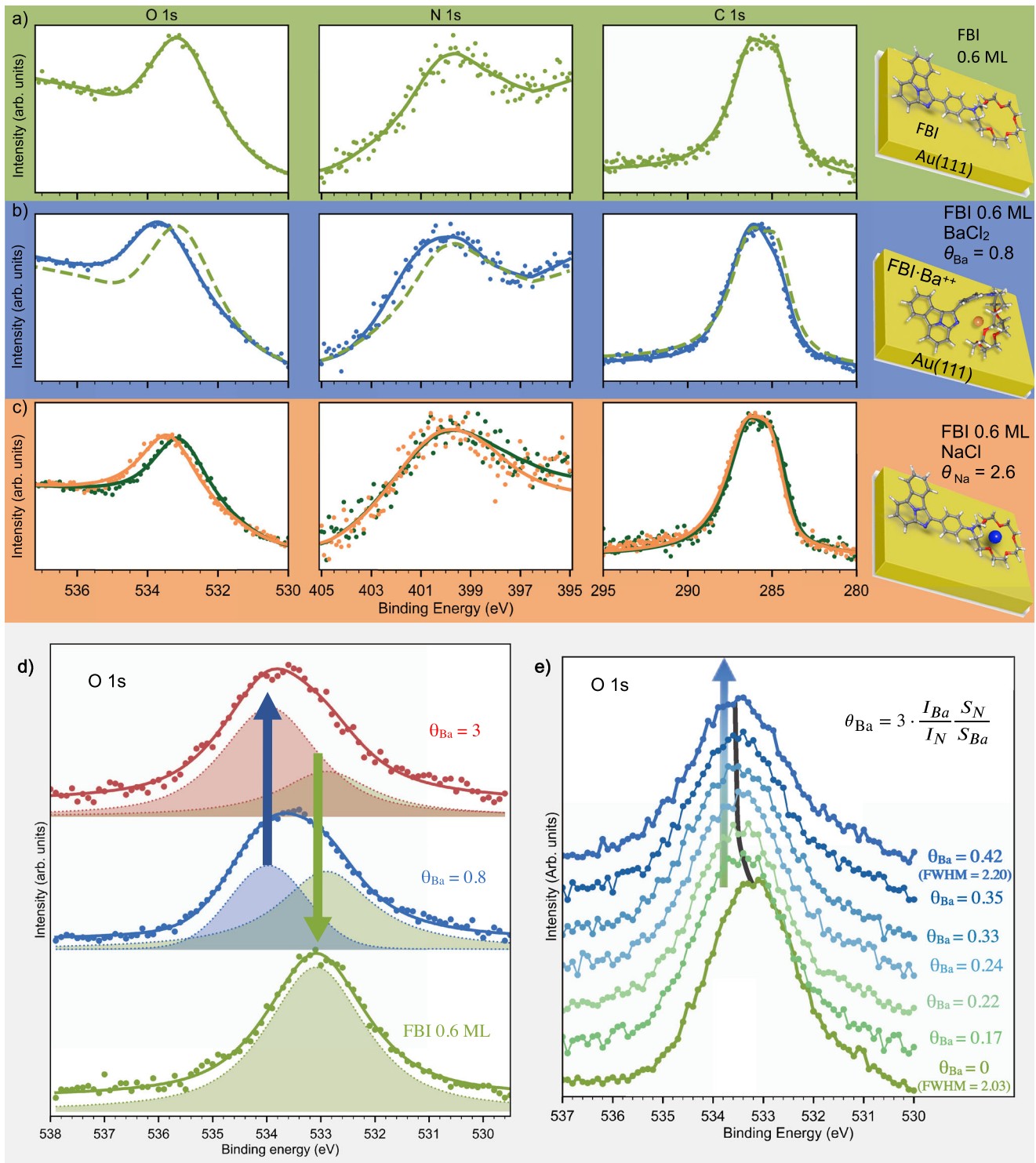

**Fig. 2 | XPS demonstration of the chemical changes induced by the molecular chelation. a** O 1s, N 1s, and C 1s core level spectra measured after sublimation of 0.6 ML of FBI deposition on Au(111); **b** FBI core level spectra measured on the previous sample (dashed spectra) after chelation with 0.80 Ba$^{+2}$ ions per FBI molecule (blue spectra); **c** O 1s, N 1s, and C 1s core level spectra measured after 0.6 ML of FBI deposited on Au(111) (green spectra) and after chelation with Na$^+$ (2.60 Na$^+$ ions per FBI molecule) (orange line). For the three panels, dots correspond to raw values and solid lines to fitted curves (the fitting procedure is discussed in the Methods section). **d** Curve-component deconvolution of O 1s spectra in **a** and **b** as well as after chelation with $\theta_{Ba} = 3$. **e** Evolution of the O 1s core level as a function of the Ba$^{+2}$ deposition on 0.9 ML of FBI on Au(111). The spectra were manually shifted in the y-axis to better show the evolution. The O 1s spectra in **d** and **e** are displayed after subtraction of the contribution from Au 4p 3/2, in order to emphasize the spectral changes.

with having intact molecules. In the N 1s region, around 400.4 eV, a faint peak is visible. The position of the maximum is compatible with the enamine-imine groups of the molecular composition. Finally, the O 1s core level presents a single component peak centered at around

533.0 eV, which is compatible with previous reports on closely related crown ether groups[38]. The ratios between the core levels, C/O = 6.2, C/N = 10.3, are in agreement with having molecules of $C_{31}N_3O_5H_{35}$ stoichiometry on the surface (FBI molecule stoichiometry).

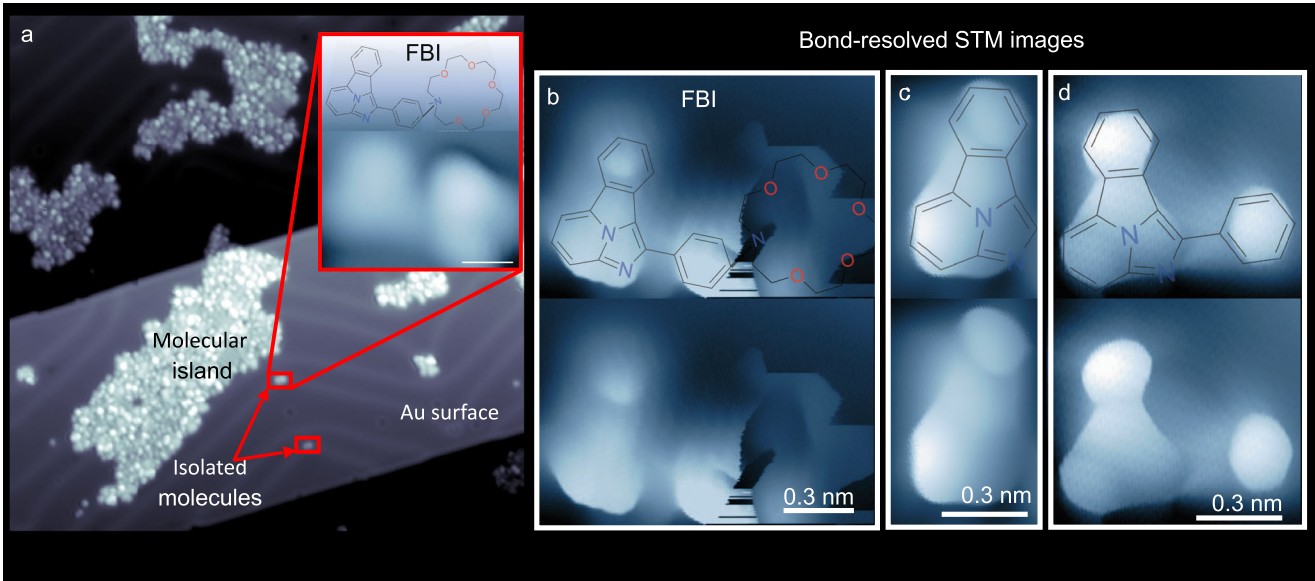

**Fig. 3 | STM images of FBI and FBI derivative molecules on Au(111). a** Large-scale STM image (50 × 50 nm²) of around 0.4 ML of FBI molecules deposited on Au(111) surface and measured at 4 K. Red squares show isolated molecules. Inset: constant current zoom STM image of a FBI molecule (I = 60 pA / U = 1.4 V) deposited on Au(111). **b–d** Bond-resolved STM images measured with a CO-functionalized probe (constant height, U = 5 mV) of individual FBI molecules: **b** benzo[*a*]imidazo[5,1,2-*cd*]indolizine fluorophore with phenyl ring and aza-crown ether (exactly the same molecule shown in the inset in **a**); **c** benzo[*a*]imidazo[5,1,2-*cd*]indolizine fluorophore; **d** benzo[*a*]imidazo[5,1,2-*cd*]indolizine fluorophore with phenyl ring but without the crown ether. For clarity, the same image is shown twice, top one with and bottom one without the molecular model superimposed to guide the eyes.

In addition to the XPS analysis, we have confirmed the intact sublimation of the molecules by STM. Deposition of a submonolayer of FBI mostly leads in disordered islands coexisting with isolated molecules (Fig. 3a). The lack of chemical order on the islands makes it difficult to identify specific molecules, and interpretation depends on the viewer. However, it is feasible to spot characteristics that mirror the features of such molecules (some examples are illustrated in Supplementary Fig. 1). With this, molecular fragmentation at the surface or in the evaporation process is ruled out. STM images of single molecules (inset in Fig. 3a) are identified by two main triangular-like lobes. Based on the molecular structure, one of them should correspond to the fluorophore and the other to the crown ether moiety bonded to the phenyl ring. The apparent height of both is different, with one of them appearing lower and flatter than the other.

In order to correlate the apparent molecular shape and the molecular structure, bond-resolved STM images were taken. For that, the STM tip was functionalized with CO molecules operating in the repulsive tip-sample interaction regime, a method extensively used for characterizing planar organic molecules[39,40]. Application of this measurement mode to the FBI molecule results in the images displayed in Fig. 3b. For eye guidance, the STM image has been shown twice, one with and one without the superimposed molecular model. An internal bonding structure close to the fluorophore structure is resolved on the left region of the molecule, while the aza-crown ether region is not well resolved. Due to its larger conformational flexibility and its non-planarity, tip stability is not good enough and, as a consequence, the contrast observed in the aza-crown ether part is an artefact related to excessively strong interactions with the flexible CO tip-apex[41,42]. To confirm the imaging signature of the fluorophore, we also sublimated two FBI derivatives specifically synthesized without the aza-crown ether component. Figure 3c and d shows the bond-resolving images of FBI derivatives sublimated on Au(111) surface: the fluorophore (Fig. 3c) and the fluorophore with the phenyl ring but without the aza-crown ether (Fig. 3d). The absence of the aza-crown ether allowed the molecules on the surface to adopt a planar, rigid conformation, which clearly improves the image acquisition, with the carbon framework of

the fluorophore well visualized. It is remarkable that none of these FBI derivatives forms molecular islands. Instead, only dimers or molecular tetramers are distinguished in the STM images. This supports the previous assessment, that FBI molecules remain intact (non fragmented) after sublimation.

## Chemical demonstration of chelation

Once the presence of intact FBI molecule on the surface was confirmed, $BaCl_2$ was sublimated to test the molecular chelation. Prior to the sublimation on the FBI, $BaCl_2$ was sublimated on clean Au(111) surface to confirm its stoichiometry. By comparing the core level intensities of Ba 3*d* and Cl 2*p* core levels (taking into account the corresponding sensitivity factor of each element), the ratio Ba:Cl was 1:2 as expected. Surprisingly, when $BaCl_2$ was deposited on FBI-functionalized Au(111), this ratio changes, being around 1:1.5 ± 0.2, meaning that there is a partial desorption of chlorine atoms when $BaCl_2$ reached the sample. From the simulations we have performed on the way that $BaCl_2$ interacts with the FBI molecule (see details of the calculations in Supplementary Note 2) we conclude that the chlorine atoms behave as passive spectators in the chelation. When the FBI molecule is chelated with $BaCl_2$ salt, the coordination pattern (N · $Ba^{+2}$, Ph · $Ba^{+2}$ and crown ether · $Ba^{+2}$) is the same as the one calculated in ref. 15 and the Cl atoms play no role (see Supplementary Fig. 2). As a result, when the $BaCl_2$ molecules are placed on the FBI functionalized surface, the $BaCl_2$ molecules split, releasing Cl atoms. The fluctuations in the Ba:Cl ratio are due to partial desorption of Cl atoms from the surface. In any case, the Ba 3*d* core level does not show any significant shift when comparing the sublimation of $BaCl_2$ on bare or functionalized gold surface (as shown in Supplementary Fig. 3), suggesting that there is no change in the barium chemical oxidation state[43]. Because of this lack of stoichiometry, we refrain from using $BaCl_2$ ML to refer to the amount of sublimated molecules and, instead, we will refer to the $Ba^{+2}$ ions per FBI molecule. The $Ba^{+2}$ dose, $\theta_{Ba}$, was estimated using the ratio between the XPS intensities of the Ba 3*d* and N 1*s* core levels divided by the corresponding sensitivity factor, and always taking into account that there are 3 N atoms per molecule. The details of the ion dose calculations are included in Methods Section.

After sublimation of 0.80 $Ba^{+2}$ per FBI molecule (Fig. 2b) the molecular core levels slightly shift. Even though the three core levels shifted toward higher binding energies, the magnitude of each shift is different, being the highest for O 1$s$ (upward shift of 0.5 eV). Since each shift is different in size, there must be a chemical change and not charge doping (associated to non reacted ions), which would show a rigid shift of all core levels. According to the O 1$s$ core level curve fitting, the peak width grows and a new component has to be introduced at a higher BE position, as shown in Fig. 2d. The new component, added to the non-chelated FBI component at 532.95 eV, is centered at around 533.9 eV, position consistent with O-Ba interaction (chelated crown ether)[38]. This new component remains unshifted even for $\theta_{Ba}$ above one $Ba^{+2}$ ion per FBI molecule, as can be seen in Fig. 2c, where the $\theta_{Ba}$ was increased up to 3 $Ba^{+2}$ per FBI molecule. Figure 2e shows the evolution of the O 1$s$ for incremental amounts of $Ba^{+2}$ ions, from 0 to almost 0.5 $Ba^{+2}$ ions per FBI molecule, on 0.9 ML of FBI on Au(111). There is a monotonous shift toward higher binding energies of the maximum of the spectra. Once $BaCl_2$ is added to the sample, the O 1$s$ peak width increments by 9% FWHM and, again, a new component at around 533.9 eV is necessary to properly fit the core level spectrum. This new component remains constant in energy, and only its intensity changes (the fit is included in Supplementary Fig. 3). However, quantification is complicated because of the close proximity of the right tail of the Au $4p_{3/2}$ core level (as shown in Supplementary Fig. 3). The O 1$s$ spectra shown in Fig. 2d and e were plotted after subtracting the contribution of the Au $4p_{3/2}$ core level in order to emphasize the spectral changes.

It is important to note at this point that, even when the number of Ba ions per molecule permitted it, we never observed a complete chelation of the FBI. It is challenging to determine whether this is due to ineffective chelation or because of the designed experiment, as the latter certainly limits it. To demonstrate the $Ba^{+2}$ trapping by FBI molecule the XPS experiment needs a coverage below or close to the monolayer to avoid interference from molecular stacking (FBI molecules forming multilayer structures). Under these conditions, sublimated $BaCl_2$ can still form islands on bare gold regions (as we have seen in the STM images). This can prevent them from participating in the chelation process.

As previously discussed, the calculations determine that Ba-chelated conformation is maintained regardless of the ion source (naked $Ba^{+2}$ cation or $BaCl_2$ salt). In order to simulate this final conformation, the $BaCl_2$ molecules were initially positioned on the periphery of the crown ether rather than close to the center. Because the crown ether has both concave and convex sides, we have also performed a simulation where the barium cation was forced to interact with the sensor through the convex face. However, the most stable conformation is that in which $Ba^{+2}$ is trapped in the concave part (fully optimized geometry shown in Supplementary Fig. 2). We investigated this possibility because this coordination mode appears plausible on surfaces. In both cases, the ions starting in a lateral position with respect to the crown ether can find a metastable conformation with all the O-$Ba^{+2}$ equidistant. These calculations rule out two possible causes of the O 1$s$ core level shift: chelation of molecules at the edges of the $BaCl_2$ islands, and interactions between $BaCl_2$ ($Ba^{+2}$) and the periphery of the crown ether. Therefore, in terms of XPS, it seems reasonable to interpret the O 1$s$ core level shift as a chelation fingerprint.

To evaluate whether the nature of the ions has any influence on the oxygen core level shift for the FBI molecules, we tested the chelation with $Na^+$. We selected this ion because it was also the choice of the only prior study in which crown ether was chelated on surfaces[38]. Figure 2c shows the O 1$s$, N 1$s$ and C 1$s$ core levels measured on 0.6 ML of FBI on Au(111) before and after the sublimation of $\theta_{Na}$ = 2.60 per FBI ($\theta_{Na}$ is computed analogously to $\theta_{Ba}$). The maximum of the O 1$s$ core level shifts again for $Na^+$-chelated FBI molecules 0.5 eV toward higher BE, while there is no apparent shift on the N 1$s$ (nor C 1$s$) core level. This

results reveals that N 1$s$ is not as involved in the ion trapping as before, which agrees with the theoretical calculations where almost no FBI distortion is predicted upon chelation with $Na^+$ (as shown in Supplementary Fig. 4). This difference between $Ba^{+2}$- and $Na^+$-chelation can be interpreted as a first proof that the molecular conformation of FBI and the cation-phenyl interaction varies depending on the nature of the trapped ions.

## Molecular structural rearrangement induced by chelation

In order to visualize the structural changes undergone by the FBI molecules upon chelation, STM experiments were carried out again. Figure 4a shows the schematic representation of the most stable conformation calculated for the molecules in gas phase before and after chelation with $Na^+$ and $Ba^{+2}$ (calculation details in Supplementary Note 4). In all cases the crown ether ring is out of the plane with respect to the fluorophore, and in the case of the $Ba^{+2}$-chelated molecule, this group is folded over the fluorophore. For molecules lying on a surface, we expect to have the fluorophore parallel to the substrate, and the crown ether region out of the plane, as corroborate the STM images (Fig. 4b–d). In the three images, the fluorophore and the aza-crown ether region are distinguished. In the case of $Na^+$-chelated FBI identification is easier than for $Ba^{+2}$-chelated because the image remains almost unchanged compared to the non-chelated FBI. FBI chelated with $Ba^{+2}$ has a more complex shape. Interpretation of constant current mode STM images is always difficult because of the convolution of topographical and electronic contributions. To reduce as much as possible the influence of the latter, the images included in Fig. 4b–d were measured using bias voltages inside the molecular gap. Taking into account that the three STM images are plotted with a common color code, it can be directly observed that the apparent heights of the crown ether follow the expected trend (higher for $Ba^{+2}$-chelated molecule). Indeed, the apparent height for each molecule, as measured on the highest point of the crown, is 2.5 ± 0.1 Å, 2.1 ± 0.1 Å, and 2.9 ± 0.1 Å, for the pristine FBI and the $Na^+$- and $Ba^{+2}$-chelated molecules respectively. Thus, complexation of crown ethers with alkali metals like $Na^+$ causes the oxygen atoms to point to the center, forcing the ring to adopt a flatter conformation relative to the native molecule. Instead, the FBI molecule adopts a more three-dimensional conformation upon chelation with $Ba^{+2}$. It is worth mentioning that the aspect of these STM images resembles the electron density simulations associated with the most stable conformers shown in Fig. 4a (electron density simulations shown in Supplementary Fig. 4). To establish a more quantitative relationship between apparent molecular structural modification and chelation, scanning tunneling spectroscopy (STS) measurements were performed.

STS permits to map the local density of states and to visualize the orbitals around the Fermi level, i.e., the Lowest Unoccupied Molecular Orbital (LUMO) and the Highest Occupied Molecular Orbital (HOMO). Calculations done by DFT in ref. 15 reveal that these frontier orbitals change their relative distances upon chelation (the HOMO-LUMO gap is higher for $Ba^{+2}$-chelated molecules than for native ones). Moreover, for both unchelated and $Ba^{+2}$-chelated FBI molecules, these orbitals are mainly located at the fluorophore region (the benzoimidazoindolizine group). This is an advantage because we can scan the local density of states on the fluorophore region by STS and correlate the spectral changes with the variation in the electronic structure of the whole molecule. Thus, Fig. 4e shows the associated STS measured for the three molecules of Fig. 4b–d in the fluorophore region. The FBI molecule deposited on Au(111) has a HOMO-LUMO gap of 3.17 eV. Complexation with $Na^+$ slightly changes this band gap, while complexation with $Ba^{+2}$ increases it up to 3.63 eV.

The DFT calculations[15] also show that the lowest fluorescence emission peak of the non-chelated FBI molecule mainly comes from the de-excitation of electrons from the LUMO to both HOMO and HOMO-1. For $Ba^{+2}$-chelated molecules, the torsion of the phenyl group decreases the effective $\pi$-conjugation and promotes the LUMO-HOMO

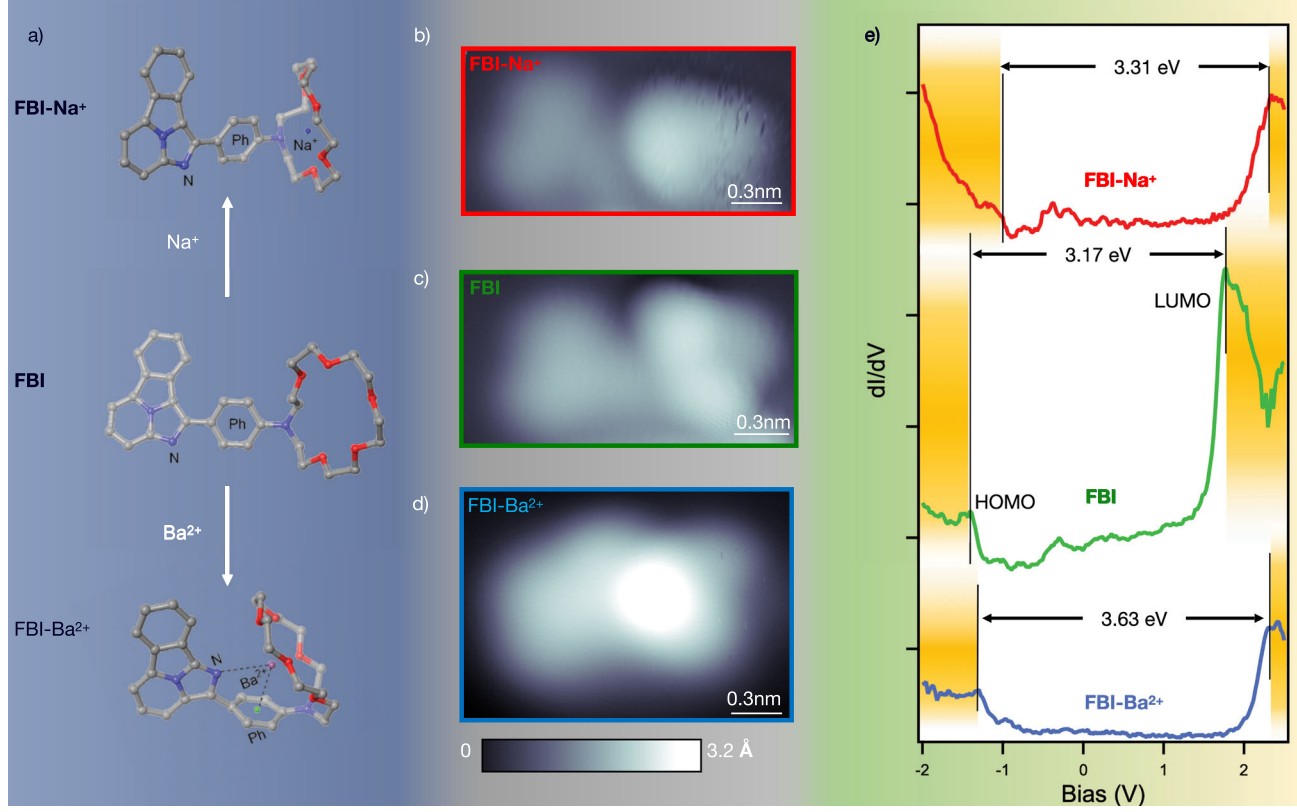

**Fig. 4 | Molecular chelation probed by STM/STS. a** Representation of the most energetically stable conformation calculated for the molecules in gas phase. **b–d** STM images of **b** Na⁺-complexed (U = 0.9 V, I = 60 pA), **c** native (U = 1.4 V, I = 60 pA), and **d** Ba⁺²-complexed (U = −0.5 V, I = 20 pA) FBI. (Scale bars = 0.3 nm).

**e** STS spectra of pristine (green), Na⁺-complexed (red), and Ba⁺²-complexed FBI (blue)FBI. Extra spectra for unchelated and chelated molecules are shown in Supplementary Fig. 5.

transition over the LUMO-HOMO-1 as the lowest emission channel. By STS we observed a clear variation in the electronic gap upon chelation with $Ba^{+2}$: the LUMO-HOMO gap increases, which explains the blue-shift fluorescence emission spectra measured in solution (Table 1).

For $Na^+$, while there is a small variation of the electronic gap, the molecule remains planar. This planarity of the molecule after chelation does not modify the $\pi$-conjugation, which is still extended into the phenyl ring. Therefore, the LUMO-HOMO distance does not significantly change. However, this small shift seems to be in contradiction with the fluorescence measurements, where no shift was measured for $Na^+$-chelated FBI molecules compared to unchelated FBI. While the magnitude of the HOMO-LUMO gap defines the maximum emission energy, the closely spaced vibrational energy levels in the ground state result in a distribution of photon energies. As a result, fluorescence is normally observed as intensity over a band of wavelengths rather than a sharp line. Hence, the small gap modification we have measured by STS could produce a correspondingly small shift in the emission spectra. However, this shift would be undetectable because of the broadness of the spectra. Table 1 summarizes the absorption and fluorescence emission maxima $\lambda_{max}^{emi}$ and $\lambda_{max}^{abs}$ associated to the transition of electrons

from HOMO to LUMO+n states (absorption), as well as from LUMO, to HOMO-n (fluorescence).Unfortunately, because of the metallic character of the substrate we cannot measure the fluorescence emission directly on gold (the excitation energy is dumped to the metal without further emission).

### Chelation tested on different surface supports

Finally, we have observed that chelation also takes place on other surfaces, in particular Cu(111) and Indium Tin Oxide (ITO). We chose Cu(111) because it is a more reactive substrate. This enabled us to study whether the molecule-substrate interaction could alter the chelation response. On the other hand, ITO was selected as a promising candidate for the potential implementation of a Barium tagging detector on a xenon-based TPC[15]. As a degenerated semiconductor, ITO is transparent, which would allow the direct detection of fluorescence in transmission. Furthermore, its conductivity is high enough to guide the $Ba^{+2}$ ions towards the sensor surface, thus facilitating their capture by FBI. On the other side, the conductivity of ITO may be potentially low enough to avoid the fluorescence quenching that is expected at conductive surfaces.

As previously discussed, we focus again on the O 1s core level to see whether the conclusions extracted before about its shift apply also on these surfaces and we can use it as a fingerprint of the chelation. Figure 5 shows the O 1s core level of FBI on Cu(111) and ITO. We used a $Ba^{+2}$ dosing of $\theta_{Ba} = 10$ for 0.6 ML of FBI on Cu(111) and $\theta_{Ba} = 17.5$ for 0.7 ML of FBI on ITO. In both cases, we observed the chemical shift on the O 1s towards higher BE, indicating that chelation is happening. In the case of Cu(111), the shift is about 0.7 eV, while for ITO it is around 0.9 eV. As far as ion capture is concerned, FBI chelation of $Na^+$ on Cu(111) was also tested with $\theta_{Na} = 3.6$. Again, O 1s shows an upwards shift associated with chelation (inset in Fig. 5a). Supplementary Figure 6 shows that

### Table 1 | Comparison of STS gap values and absorption and fluorescence spectroscopy

| Species | Band gap / eV (nm) | $\lambda_{max}^{abs}$ / nm | $\lambda_{max}^{emi}$ /nm |
|---|---|---|---|
| FBI | 3.17 (391.1) | 432.5 | 489 |
| FBI-Na⁺ | 3.31 (374.6) | 430 | 489 |
| FBI-Ba⁺² | 3.63 (341.6) | 420.5 | 428 |

Band gap values measured by STS on Au(111) vs. absorption ($\lambda_{max}^{abs}$) and fluorescence emission ($\lambda_{max}^{emi}$) spectral peaks measured in solution[15].

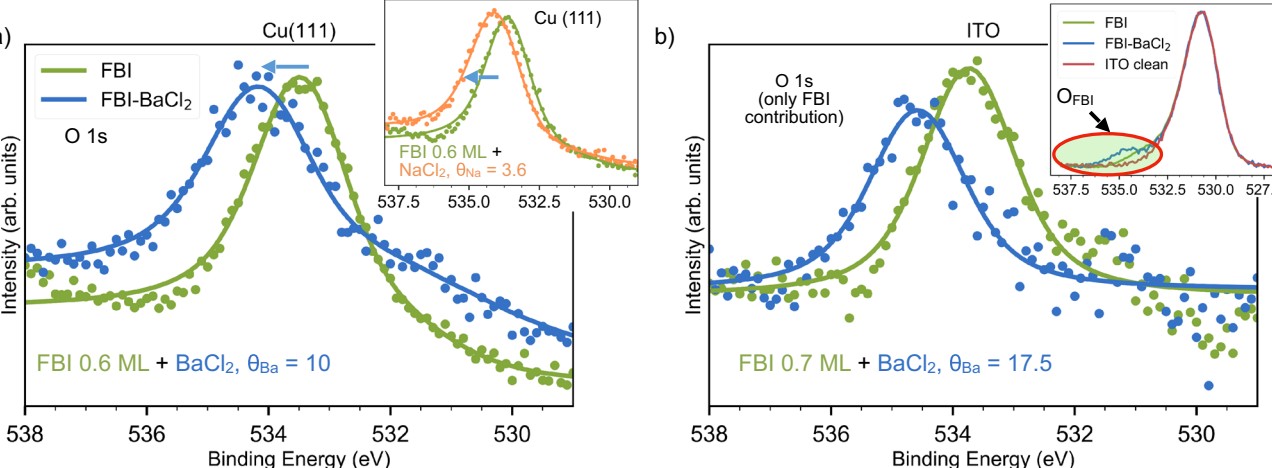

**Fig. 5 | O 1s core level evolution upon chelation measured on two sub-monolayer FBI-functionalized surfaces. a** chelation on Cu(111) sublimating BaCl$_2$ and NaCl (inset); **b** chelation on ITO. In this case, the substrate component (O$_{ITO}$) was subtracted to emphasize the O 1s contribution coming from FBI (O$_{FBI}$). The entire spectra are shown inserted in **b**. Dot spectra correspond to raw values and solid lines to fitted curves.

the O 1s shifts for low $\theta_{Ba}$ doses on Cu(111), with a similar shift evolution as the one observed on Au(111) for low $\theta_{Ba}$ doses (in Fig. 2d).

Quantifying the exact values of the O 1s core level shift and chelation efficiency was not possible. Notice that in the case of Cu(111), when the FBI-functionalized surface is exposed to BaCl$_2$, the residual contamination partially oxidizes the surface. For this reason, the core level has a smaller component at lower BE, around 531 eV (see Supplementary Fig. 6), associated to Cu$_2$O[44]. On the other hand, because of the presence of oxygen in the ITO structure, the analysis of the O 1s core level shifts required a subtraction of the core level measured on the bare ITO to highlight the FBI and Ba$^{+2}$ chelated FBI contributions (O$_{FBI}$). The result of the subtraction is shown in Fig. 5b, and the inset gathers the original normalized spectra, including the contribution to the O 1s coming from the ITO (O$_{ITO}$).

These results ensure that chelation is independent from the choice of substrate, even though the values of the shifts are different for the three substrates. In photoemission, the absolute value of the core level shift depends on many factors such as the substrate, the molecular coverage, the presence of defects or other molecules. However, what is important here is that in the three cases the core level shifts in the same direction and with similar magnitude.

To summarize, by combination of highly sensitive surface science techniques, we have unraveled the chemical and conformational changes that occur upon chelation of FBI molecules. We started with Au(111), a well-known surface that served as model, and moved on toward more suitable surfaces. Furthermore, the changes are in agreement with the calculations for the behavior of free standing molecules. Regarding the bicolor property of the sensor, fluorescence emission could not be directly measured because of the metallic character of the substrates. Nonetheless, the measured variations in the molecular HOMO-LUMO gap are consistent with the observed emission shift for Ba$^{2+}$-chelated molecules. Moreover, they are also consistent with the absence of such shift, observed and predicted for other ions, such as Na$^+$.

The demonstration of chelation in vacuum of FBIs by Ba$^{+2}$ ions on surfaces in submonolayer regime is a major step towards the development of a sensor capable of single barium tagging. This sensor would have the immediate application in a new, ultra-low background xenon-based $\beta\beta0\nu$ detection experiment. Furthermore, this study has also important implications beyond the field of particle physics. Aza-crown ether groups have demonstrated to chelate alkali ions with high affinity in solution. Here we showed the capability of these molecules to trap big or small ions (Ba$^{+2}$ or Na$^+$) on surfaces too. This could have

important applications in drug carriers[45], photo-switching devices[46,47], or different types of ion sensors.

## Methods

The experiments were performed in two different UHV chambers, one for XPS experiments and the other for STM experiments. Both chambers have a base pressure of $1 \times 10^{-10}$ mbar. Prior to molecular deposition, the Au(111) and Cu(111) and Indium Tin Oxide (ITO) surfaces were cleaned via cycles of Ar sputtering and annealing to 500 °C and their cleanliness was checked by XPS. The agreement between experiments performed on different substrates and subsequent preparations confirmed the reproducibility of the evaporation and chelation processes.

### Molecular evaporation

Pure FBI, BaCl$_2$, and NaCl powders were evaporated from homemade Knudsen cell evaporators. To avoid cross-talk between FBI and the salts and to exclude any possibility of chelation of the molecule inside the cell, the FBI and BaCl$_2$(NaCl) evaporators were located in two separated parts of the UHV chambers. To ensure sublimation reproducibility, each molecular evaporator cell (containing the FBI, BaCl$_2$, and NaCl) was removed from the XPS chamber and installed in the STM chamber.

FBI molecules were synthesized using the procedure described in ref. 15, while BaCl$_2$ and NaCl were commercially (Sigma–Aldrich) available. The molecules were used after degassing in UHV. The molecular evaporation rate was monitorized using a quartz microbalance. Afterwards, the amount of molecules on the surfaces was quantified by analyzing the relative intensity of the XPS core level peaks and the percentage of covered surface in STM. FBI evaporation was very stable and very reproducible, since the same rate for the same sublimation temperature was monitored. This reproducibility rules out fragmentation inside the evaporation cell. In the opposite case, if the molecules broke during the evaporation process, then the evaporation fragments would sublimate at different sublimation temperatures, and the evaporation would became non-reproducible. The simpler FBI derivative molecules (without crown ether) sublimate at 60–80 °C while the intact FBI molecule sublimates at 140–150 °C.

### Absorption and emission spectra

UV-vis spectra were acquired on a Shimadzu UV-2600 Spectrophotometer. Emission spectra were acquired on an Agilent Cary Eclipse Fluorescence Spectrophotometer. Excitation and emission monochromator bandwidth was fixed at 5 nm. Spectra were recorded

at $5 \times 10^{-5}$ M solutions of FBI and equimolecular quantities of the corresponding perchlorate cation salt for the chelated species. Emission spectra were acquired using 250 nm light.

## STM experiments

STM experiments were performed with a commercial Scienta-Omicron ultra-high vacuum LT-STM at 4.3 K. A W-tip was used. Topography images were measured using constant current mode, while for bond resolution images constant height mode was used (Fig. 3b–d). During the STM experiments, the tip was functionalized with CO for bond resolution STM by exposing the sample to low pressure (-1 × 10⁻⁸ mbar) of CO whilst the sample was held below 7 K. CO molecules were trapped by the tip from their adsorption sites via scanning over them or by applying -2 V bias voltage pulses.

## XPS experiments

The XPS measurements were carried out using a Phoibos-100 electron analyzer (SPECS GmbH), using a non monochromatic Al K$\alpha$ photon source of 1486.6 eV. The spectra were calibrated to the substrates main core level (Au 4$f$, Cu 2$p$, and In 3$d$, respectively).

The evaporation thicknesses were estimated using the attenuation of the most intense substrate core levels, i.e., Au 4$f$, Cu 2$p$, and In 3$d$ for Au(111), Cu(111), and ITO, respectively. The calculations followed the guidelines provided in ref. 48. For this purpose, we estimated the electron Effective Attenuation Length (EAL) through the FBI layers for electrons with kinetic energies of 1402.6, 1041.6, and 554.6 eV (corresponding to Au $f_{5/2}$, In $3d_{5/2}$ and Cu $2p_{3/2}$ for Al K$\alpha$ radiation). The resulting EALs are 3.87, 3.05, and 1.85 nm,

respectively. The thickness of the FBI samples was then estimated using the intensity of clean substrate core level as reference and the attenuated intensity after the evaporation. To correct for systematic uncertainties affecting different sets of data, the intensities of all spectra were rescaled to the main substrate core levels (Au 4$f$, Cu 2$p$, and In 3$d$, respectively). The core level peak areas were numerically integrated between fixed local minima. This yields an error in the stoichiometry ratios of 7%.

The stoichiometry of the FBI (BaCl$_2$) was calculating following the expression $R(A/B) = (I_A/I_B \cdot S_B/S_A)$ where A and B are the different molecular elements, $I_X$ is the integrated area under the main core level and $S_X$ is the corresponding sensitivity factor of this core level.

The amount of Ba$^{+2}$(Na$^+$) per FBI molecule were estimated by computing the ratio $\theta = I_{Ba}/(I_N/3) \cdot S_N/S_{Ba}$, where $I_{Ba}$, $I_N$ are total areas of Ba 3$d$ and N 1$s$ core levels, respectively, and $S_{Ba} = 25.84$, $S_N = 1.80$ are the corresponding atomic sensitivity factors[49]. The factor 3 responds to stoichiometric reasons, considering 3 N atoms per molecule.

## XPS fitting

The spectra fitting was performed using custom-made software written in Python and the lmfit library[50]. An example of fit can be seen in Fig. 6. The core levels were modeled as a pseudo-Voigt function (Lorentzian to Gaussian ratio of 0.6) convoluted with a Shirley type background. The error in the estimation of the position of the maxima derived from fitting is 50 meV. However we only consider as significant shifts higher than 100 meV, which is the energy step we use in the adquisition. The goodness of fit is given by a $\chi^2$ parameter of about 0.05 counts (in A.U.) when the data are normalized to a maximum intensity of 1. Here $\chi^2 = \sum_i (y_i - Y_i)^2/Y_i$, where $y_i$ and $Y_i$ are the data intensity values and the expected values from the fit, respectively.

## Data availability

The original XPS core level data presented in this study have been deposited in the Source Data file. The STM data that support the findings of this study are available from the corresponding author upon request. Source data are provided with this paper.

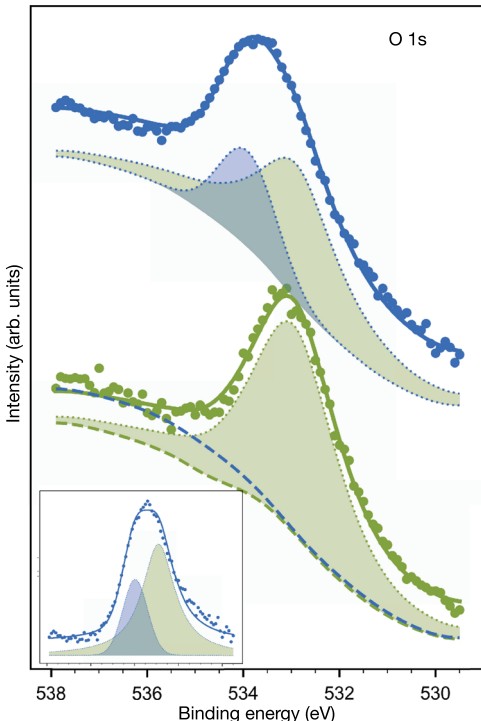

**Fig. 6 | Curve deconvolution for O 1$s$ in FBI 0.6 ML on Au(111) before and after chelation.** Detail of fitting components (dotted lines) for O 1$s$ in FBI 0.6 ML (green) and FBI 0.6 ML + BaCl$_2$, $\theta = 0.8$ (blue) from Fig. 2a and b. Circles and solid lines represent raw data and best fit respectively. The green component was fitted to the unchelated FBI data (green circles) at 532.95 ± 0.05 eV, with width 2.16 ± 0.04 eV at FWHM. This component was then fixed in position and width for the chelated FBI data (blue dots). An additional component was fitted to the chelated FBI CL, shown as blue filled area, at 533.91 ± 0.05 eV, with 1.41 ± 0.05. These values were obtained from the fit as free parameters. The background of each curve is shown as dashed lines in green and blue respectively. Inset: same blue curve and components with its background subtracted.

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

## Acknowledgements

This material is based upon work supported by the following agencies and institutions: the European Research Council (ERC) under ERC-2020-SyG 951281; the MCIN/AEI/10.13039/501100011033 of Spain and ERDF A way of making Europe under grants PID2020-114252GB-I00, PID2019-107338RB-C63, PID2019-104772GB-I00, PID2019-111281GB-I00, and RTI2018-095979, the Severo Ochoa Program grant CEX2018-000867-S; the Basque Government (GV/EJ) under grants IT-1553-22, IT-1591-22. The NEXT Collaboration acknowledges support from the following agencies and institutions: the European Union's Framework Programme for Research and Innovation Horizon 2020 (2014-2020) under Grant Agreement No. 957202-HIDDEN; the MCIN/AEI of Spain and ERDF A way of making Europe under grants RTI2018-095979 and PID2021-125475NB, the Severo Ochoa Program grant CEX2018-000867-S and the Ramón y Cajal program grant RYC-2015-18820; the Generalitat Valenciana of Spain under grants PROMETEO/2021/087 and CIDEGENT/2019/049; the Department of Education of the Basque Government of Spain under the predoctoral training program non-doctoral research personnel; the Portuguese FCT under project UID/FIS/04559/2020 to fund the activities of LIBPhys-UC; the Pazy Foundation (Israel) under grants 877040 and 877041; the US Department of Energy under contracts number DE-AC02-06CH11357 (Argonne National Laboratory), DE-AC02-07CH11359 (Fermi National Accelerator Laboratory), DE-FG02-13ER42020 (Texas A&M), DE-SC0019054 (Texas Arlington) and DE-SC0019223 (Texas Arlington); the US National Science Foundation under award number

NSF CHE 2004111; the Robert A Welch Foundation under award number Y-2031-20200401. Finally, we are grateful to the Laboratorio Subterráneo de Canfranc for hosting and supporting the NEXT experiment.

## Author contributions

J.J.G.-C., F.P.C., and C.R. conceived the project. C.R. designed and coordinated the experiments and data analysis. P.H.G. and M.I. performed and analyzed the XPS experiments. J.P.C., A.B.L., T.W., D.G.O., and M.C. performed the STM experiments. I.R. and B.A. carried out the chemical synthesis. R.G.M., A.A., and Z.F. performed the characterization in solution of fluorescence studies. F.P.C. performed the DFT calculations. P.H.G., J.J.G.-C., F.P.C., and C.R. wrote the manuscript. F.M. and the NEXT collaboration assisted with editing and revision of the manuscript.

## Competing interests

The authors declare no competing interests.

## Additional information

## NEXT collaboration

C. Adams[6], H. Almazán[7], V. Álvarez[8], B. Aparicio[4], A. I. Aranburu[5], L. Arazi[9], I. J. Arnquist[10], S. Ayet[11], C. D. R. Azevedo[12], K. Bailey[6], F. Ballester[8], J. M. Benlloch-Rodríguez[2], F. I. G. M. Borges[13], S. Bounasser[7], N. Byrnes[14], S. Cárcel[15], J. V. Carrión[15], S. Cebrián[16], E. Church[10], C. A. N. Conde[13], T. Contreras[17], F. P. Cossío [ID][2,4], A. A. Denisenko[18], E. Dey[18], G. Díaz[19], T. Dickel[11], J. Escada[13], R. Esteve[8], A. Fahs[7], R. Felkai[9], L. M. P. Fernandes[20], P. Ferrario[2,3], A. L. Ferreira[12], F. W. Foss[18], E. D. C. Freitas[20], Z. Freixa[5], J. Generowicz[2], A. Goldschmidt[21], J. J. Gómez-Cadenas [ID][2,3], R. González-Moreno[1], R. Guenette[7], J. Haefner[17], K. Hafidi[6], J. Hauptman[22], C. A. O. Henriques[20], J. A. Hernando Morata[19], P. Herrero-Gómez[1,2], V. Herrero[8], J. Ho[7], P. Ho[18], Y. Ifergan[9], B. J. P. Jones[14], M. Kekic[19], L. Labarga[23], L. Larizgoitia[2], P. Lebrun[24], D. Lopez Gutierrez[7], N. López-March[8], R. Madigan[18], R. D. P. Mano[20], J. Martín-Albo[15], G. Martínez-Lema[9], M. Martínez-Vara[2,15], Z. E. Meziani[6], R. Miller[18], K. Mistry[14], F. Monrabal[2,3], C. M. B. Monteiro[20], F. J. Mora[8], J. Muñoz Vidal[15], K. Navarro[14], P. Novella[15], A. Nuñez[2], D. R. Nygren[14], E. Oblak[2], M. Odriozola-Gimeno[2], B. Palmeiro[7], A. Para[24], M. Querol[15], A. B. Redwine[9], J. Renner[19], L. Ripoll[25], I. Rivilla [ID][2,3,4], J. Rodríguez[8], C. Rogero [ID][1,2][✉], L. Rogers[6], B. Romeo[2,26], C. Romo-Luque[15], F. P. Santos[13], J. M. F. dos Santos[20], A. Simón[9], M. Sorel[15], C. Stanford[7], J. M. R. Teixeira[20], J. F. Toledo[8], J. Torrent[2,25], A. Usón[15], J. F. C. A. Veloso[12], T. T. Vuong[18], J. Waiton[7] & J. T. White[27]

[6]Argonne National Laboratory, Argonne, IL 60439, USA. [7]Department of Physics and Astronomy, Manchester University, Manchester M13 9PL, UK. [8]Instituto de Instrumentación para Imagen Molecular I3M (CSIC-UPV), Valencia E-46022, Spain. [9]Unit of Nuclear Engineering, Faculty of Engineering Sciences, Ben-Gurion University of the Negev, Beer-Sheva 8410501, Israel. [10]Pacific Northwest National Laboratory (PNNL), Richland, WA 99352, USA. [11]II. Physikalisches Institut, Justus-Liebig-Universitat Giessen, Giessen, Germany. [12]Institute of Nanostructures, Nanomodelling and Nanofabrication (i3N), Universidade de Aveiro, Campus de Santiago, Aveiro 3810-193, Portugal. [13]LIP, Department of Physics, University of Coimbra, Coimbra 3004-516, Portugal. [14]Department of Physics, University of Texas at Arlington, Arlington, TX 76019, USA. [15]Instituto de Física Corpuscular (IFIC), CSIC & Universitat de Val`encia, Paterna E-46980, Spain. [16]Centro de Astropartículas y Física de Altas Energías (CAPA), Universidad de Zaragoza, Zaragoza E-50009, Spain. [17]Department of Physics, Harvard University, Cambridge, MA 02138, USA. [18]Department of Chemistry and Biochemistry, University of Texas at Arlington, Arlington, TX 76019, USA. [19]Instituto Gallego de Física de Altas Energías, Univ. de Santiago de Compostela, Campus sur, Santiago de Compostela E-15782, Spain. [20]LIBPhys, Physics Department, University of Coimbra, Coimbra 3004-516, Portugal. [21]Lawrence Berkeley National Laboratory (LBNL), Berkeley, CA 94720, USA. [22]Department of Physics and Astronomy, Iowa State University, Ames, IA 50011-3160, USA. [23]Departamento de Física Teórica, Universidad Autónoma de Madrid, Campus de Cantoblanco, Madrid E-28049, Spain. [24]Fermi National Accelerator Laboratory, Batavia, IL 60510, USA. [25]Escola Polit`ecnica Superior, Universitat de Girona, Girona E-17071, Spain. [26]Laboratorio Subterráneo de Canfranc, Canfranc Estación E-22880, Spain. [27]Department of Physics and Astronomy, Texas A&M University, College Station, TX 77843-4242, USA.

