## [Peer Review File · Nature Communications]

Ba²⁺ ion trapping using organic submonolayer for ultra-low background neutrinoless double beta detectorREVIEWER COMMENTS

Reviewer #1 (Remarks to the Author):

The paper deals with a system and a surface reaction that are specifically interesting for the development of an experiment that aim to demonstrate that neutrinos are their own antiparticles by the occurrence of a characteristic nuclear reaction where a critical step is the detection of Ba⁺⁺ ions. I don't judge the scientific interest of the paper because it's related to a field of research that is not my own.

Concerning the surface chemistry and physics side I think that the paper is well written, clear and the conclusion are well founded on the experimental findings

I just would like to point out some issues that I think the authors should answer before publication.

At page 5 the authors claim to have demonstrated that only Ba⁺⁺ ions cause structural changes to the molecule of interest but they just have a counter example that is Na⁺ ions. I wonder if other atoms could also produce the same type of structural modification that in turn induce the shift in fluorescence frequency. We should worry only of atoms that could have some chance to be present in the environment of the actual detection experiment. Maybe the authors should comment on this point.

On page 9 the authors write that the observed chemical shift of the O1s core level induced by the deposition of BaCl₂ and NaCl is not a "just a doping shift" since there is a formation of a new O1s component in the XPS spectra. I think they should clarify what they intend as a doping shift. If they intend that it's a shift induced by the presence of metal ion that produce a local dipole field that could be still the case depending from the distribution on the surface of the ions. Alkaline metals on surface are known to produce shifts of core levels of neighbouring atoms, effect that has been interpreted as a local work function modification. Furthermore, if the molecules containing Ba ions coalesce in islands, we could still have 2 peaks in XPS.

On page 9-10 the concept that the O1s line width increases because of the formation of a second component is repeated twice.

On page 12 the authors claim that in this specific system they can detect conformational changes in STM images collected in constant current mode since there is "no variation in the electronic structure". Is not very clear to me what they mean since, clearly, there are modifications of the electronic structure that, in fact, cause the change in fluorescence and the band gap, maybe they mean that there is no variation in the electron density of states detected by the STM in the specific energy range probed in those measurements.

On page 14 they write "The absence of substantial frontier molecular orbital gap upon complexation with Na⁺". It is not very clear to me what they mean, maybe the absence of modifications in the frontier orbital?

On page 16: the authors say that they "enhance" the contributions of the molecule to the O1s peak by subtracting the substrate contribution. I don't think enhance is the write word, maybe highlight or select?

Reviewer #2 (Remarks to the Author):

What are the noteworthy results?

Chemical sensors have been proposed to detect the daughter ion (Ba²⁺) produced in double beta decay of ¹³⁶Xe. This Ba-tag in coincidence with total electron energy near the Q-value of the decay (2.458MeV) would make a particularly robust signature of neutrinoless double beta decay with zero (or at least practically zero) background. As such, realizing this Ba-tagging could significantly improve

the sensitivity of the NEXT experiment to neutrinoless double beta decay. Neutrinoless double beta decay is one of the primary ways to probe the Majorana nature of neutrinos and conservation of lepton number. Discovering if neutrinos are Majorana particles would have deep implications for neutrino physics and cosmology, thus high sensitivity techniques are particularly important. The main result presented in this paper is demonstrating that the FBI technique executed in an ultra-high vacuum environment compatible with an actual neutrinoless double beta decay experiment produces results consistent with prior tests in aqueous solution. Prior to this the FBI tagging of Ba²⁺ had only been demonstrated in aqueous solution.

Will the work be of significance to the field and related fields? How does it compare to the established literature? If the work is not original, please provide relevant references.

Yes, in this reviewer's opinion this is an important proof-of-principle on the road to realizing Ba-tagging in the full-scale experiment and I believe it will be of interest to the neutrinoless double beta decay community. To the best of my knowledge the work presented is original. The authors have an extensive bibliography cited related prior work

Does the work support the conclusions and claims, or is additional evidence needed?

I think the evidence presented is reasonable for the conclusions drawn. Although, I think the curves in Fig 4-e are mislabeled/mis-described in the caption. The curve with the HOMO-LUMO gap of 3.63eV (green) is the FBI-Ba²⁺ right? The shift from 3.17eV for just FBI to 3.63eV for FBI-Ba²⁺ is the key result consistent with the blue shift of fluorescence observed for earlier studies in solution (Table 1). So please check/correct the caption and coloring of Fig 4.

This comment is beyond the scope of the paper which aims to demonstrate the FBI-Ba²⁺ fluorescence shift in UHV, and so should not be interpreted as a criticism of this paper but it would be important in later studies to investigate if other species could result in a shift close to the Ba²⁺ shift, this paper shows that FBI-Na⁺ results in a different fluorescence shift. What about, for example, the daughter of 110mAg decay?

Are there any flaws in the data analysis, interpretation and conclusions? Do these prohibit publication or require revision?

Just my earlier comment, which I think is just a mislabeling in the caption of Fig 4. The feature associated with the HOMO level of the red curve of Fig 4e is a little difficult to see. Perhaps this could be better emphasized in the figure.

Is the methodology sound? Does the work meet the expected standards in your field?
To this reviewer, yes.

Is there enough detail provided in the methods for the work to be reproduced?

To this reviewer yes, though I would point out I'm not an expert in the practical details of executing XPS/STM measurements so I would defer this question to reviewers with this experience.

Some other comments:

Line 64 (abstract) suggest to spell out FBI

Line 87, ¹³⁰Te is the main isotope under investigation in the Te experiments suggest to mention that instead of ¹²⁸Te, also suggest to reference the Te experiments (CUORE and SNO+)

Line 185, since only Ba⁺⁺ and Na⁺ were investigated, I feel the statement "We demonstrate that only Ba²⁺ ions induce" is too strong. Suggest "Of the species investigated only Ba²⁺ ..."

Line 229 typo structural  structurally

Line 365 suggest to spell out binding energy (BE) apologies if I missed it earlier

Observation: Figure 2 was difficult for me to read on the printed version but fine in the electronic version

Line 439: I think I know what you mean, but I felt "a real chemical change" is rather loosely defined, suggest to rework the message of that sentence.

Table 1: Suggest to repeat in the caption the reference(s) for the absorption/emission measurements in solution results listed.

Line 633 typo on substrate

Fig. 5 caption got a little garbled in the pdf rendering I received

Line 671 suggest to define ϕ (it is defined in methods but could be helpful to define it here or forward reference to methods)

Suggestion: If available, some comment in the conclusions about the reproducibility could be helpful, for example how many samples were tested ? Were possible aging effects investigated ?

Reviewer #3 (Remarks to the Author):

This manuscript addresses the adsorption of a crown ether derivative (FBI) on distinct supports (mainly Au(111), but also Cu(111) and ITO) in ultra-high vacuum and discusses the trapping of Ba and Na ions by these single-layer organic films. Based on X-ray photoelectron spectroscopy (XPS) and high-resolution scanning tunnelling microscopy (STM), the authors conclude FBI adsorbs intact on Au(111). Shifts in the O 1s XPS signal are taken as fingerprint for chelation, with Ba inducing a considerable distortion of the crown ether, whereas Na induces minor modifications. The HOMO - LUMO gap is addressed by scanning tunnelling spectroscopy (STS) and is largest for the FBI-Ba²⁺ complex, with similar trends observed in fluorescence spectroscopy in solution.

This study and the choice of crown ether derivative, ions, and supports is motivated by the potential use as fluorescent bicolour sensor in detectors for the neutrinoless double beta decay. This approach, described in a recent Nature article by some of the authors (ref. 5), is fascinating and provides convincing justification and a unique selling point of this work, that essentially presents a surface science study. It is interesting in its own right, as crown ether derivatives are rarely studied on surfaces, with only very few reports providing single molecule characterization. Accordingly, the manuscript addresses a highly relevant topic, provides novelty, and might be appealing for the broad readership of Nature Communications.

Unfortunately, however, some of the conclusions do not seem to be proven beyond reasonable doubt. To be fully convincing and meet the standards of a high impact journal, the manuscript would require additional data and analysis, further corroborating the interpretation. Thus, the reviewer can not recommend acceptance of the manuscript in its present form for publication in Nature Communications.

Some issues and questions are listed below, following the order/subsections of the manuscript.

Regarding molecular sublimation

- XPS intensity ratios and high-resolution STM images of individual FBI molecules are used to confirm intact adsorption of the molecules on Au. However, matching XPS intensity ratios might not exclude fragmentation of the molecules on the support. The STM images show the coexistence of individual molecules and molecular aggregates. Could the structure of the molecular islands be resolved and can it be excluded they are formed by molecular fragments?

The STM analysis in Fig. 3 and Fig. 4 addresses subtle details of the appearance of the molecules and relates these to the molecular structure. In the molecular islands, quite an inhomogeneous contrast is observed. This raises the questions if the information extracted from selected single molecules is representative (see additional comment below).

Regarding chemical demonstration of chelation:

- The O 1s peak shift is used as fingerprint for the chelation of the metal ion. Why are no XPS data of the metal ions included? These could provide valuable information on the chelation, as discussed for example in ref. 36 for the Na 2p signature.

- No XPS data for complete chelation ($\phi=1$) are shown for the Au support. In Fig. 2e ϕ reaches only a value of 0.42. A data set with larger ϕ might be instructive. If selective chelation is achieved, one might expect a modification in the XPS data for $\phi>1$.

- When deconvoluting the O 1s signal, the authors assign a component at around 533.9 eV to the crown ether chelation, referring to ref. 36 that addresses Na chelation. The respective binding energy in ref. 36 however is different, as is the ion. The authors might clarify if the peak position of 533.9 eV was obtained as a free parameter in the fitting procedure or if it was fixed based on some considerations.

- In the conclusion, one reads about the important capability of the aza-crown ethers to interact with "many different ions". However, the manuscript only addresses two ions, so one might consider to adapt the wording.

Regarding molecular structural rearrangement

- Based on the STM data provided in Fig. 4, the authors discuss the conformations of the three configurations. This seems a bit speculative. Distinct differences are discernible between all three images. STM image simulations based on the proposed conformations would be clearly beneficial to support the conclusions. Furthermore, Fig. 3 shows individual molecules as well as molecular islands for FBI only, indicating different conformations occur even for chemically identical species. How characteristic are the individual STM images shown in Fig. 4 for the respective species?

- The C 1s peak shapes (Fig. 2) seem very similar for all three configurations. Would one not expect distinct differences between "flat" and "more three-dimensional" conformations, with C atoms lifted from the metallic support in the latter case?

- Additional information about the STS spectra in Fig. 4 should be provided. Do these spectra reflect a spatial average over the molecule or are they characteristic for specific positions?

- On page 12 one reads that the STM images are comparable, as they "were measured at bias voltages well inside the molecular gap". Is this really true? For example, the caption of Fig. 4d gives a bias voltage of -1.8 V. Judging from the corresponding spectrum in Fig. 4e, this is outside the molecular gap. This would affect the comparability of the apparent heights.

- On page 14, one reads that "the measured HOMO-LUMO gap values are in agreement with the $\lambda_{\text{max}}^{\text{emi}}$ spectra measured in solution", the conclusions claim they are "perfectly consistent". How do the authors reach this conclusion? Both the absolute values (for example 391.1 versus 489) and the trends (STS: 3.17 and 3.31: different, emission 489 and 489: identical) do not really match. Why do the authors expect an agreement at all? The different spectroscopic methods and the environment should impact the experimentally measured gaps.

Regarding chelation on different surface supports:

- The authors compare different supports (Au, Cu, ITO), concluding based on O 1s XPS data that chelation is "independent from the choice of the substrate". However, the Ba coverages are strikingly different for the substrates (for example $\phi=0.8$ on Au versus $\phi=17.5$ on ITO). How is such a comparison justified? Can it be excluded that "overdosing" ($\phi>1$) modifies the O 1s spectra?

- In the conclusion one reads that "...conformational changes occur upon chelation, independently of

the substrate..". Is this conclusion justified? The XPS data alone (Fig.5) do not seem to address the conformation.

Additional (minor) comments

- The role of the Cl, provided by both the BaCl₂ and the NaCl, should be discussed. Cl is described as "passive spectators". Is this assumption justified on a surface? In the method section, one reads that NaCl was used, while the text mentions sublimation of Na⁺. This should be clarified.
- On page 16, the residual contamination of Cu by O reflected in a Cu₂O contribution is mentioned. Is this related to the dosing procedure or does this reflect contamination issues with the bare Cu sample?
- Fig 4 is confusing. The three columns include three situations each. While the left column and the middle column match, the spectra in the right column are not consistent.
- In surface science, "phi" usually describes the work function. Here, it is used for coverage. One might consider adapting the notation.
- The manuscript should be carefully proofread, there are several typos, formal issues (e.g., caption Fig. 5), and a figure panel not mentioned in the text (Fig. 2e).

First of all, the authors would thank the reviewers the effort revising the manuscript. We really appreciate the positive comments and the suggestions and question that definitely help to improve the manuscript and making it more readable for broader audience. We have consider all of them and her e are summarized the answers to each of them as well as the actions performed.

Reviewer #1 (Remarks to the Author):

1. *At page 5 the authors claim to have demonstrated that only Ba⁺⁺ ions cause structural changes to the molecule of interest but they just have a counter example that is Na⁺ ions. I wonder if other atoms could also produce the same type of structural modification that in turn induce the shift in fluorescence frequency. We should worry only of atoms that could have some chance to be present in the environment of the actual detection experiment. Maybe the authors should comment on this point.*

It is true that the inclusion of the word “only” in the sentence “We demonstrate that only Ba²⁺ ions induce molecular structural changes...” leads to misunderstandings. We agree that although by this “only” we referred to the Ba versus Na ions comparison, the reading could induce to think that Ba²⁺ is the only metal ion that can promote the FBI molecular torsion, and this is not correct. In fact, in our previous article (cited as [5] at present manuscript) we demonstrated that, in solution, other atoms, such as Sr, can induce a similar fluorescence shift associated to molecular torsion. On surface, we have only tested Ba²⁺, Na²⁺ and recently Fe²⁺, and a geometrical distortion was only detected for Ba²⁺ chelation. In any case, regarding the referee concerns about whether residual metal atoms inside the NEXT chamber could eventually induce similar molecular torsion, similar fluorescence shift, and therefore could affect the specific event detection; we would like to point out here two additional aspects referred to the actual experiment.

The first aspect, concerns the background composition of the NEXT vessel. Residual gas analysis (RGA) of the vacuum composition were performed at NEXT vessel and only traces of light gases, such as H₂O, CO₂, Ar₂, or N₂, and some hydrocarbon molecules (55, 57 and 69 amu) were identified. Figure 1 shows the quadrupole mass spectra, zooming the region for masses below 70 amu.

Figure 1: Quadrupole mass spectra measured for the residual gases in the NEXT vessel.

Second, even if some ion traces, undetectable in RGA, would remain in the chamber background, the measuring method designed for the final Ba ion sensor should prevent the chelation with undesired ions. As we mention in the manuscript, the preferred substrates for the construction of the final sensor are transparent semiconductors, such as ITO. The reason for this is that this kind of high dielectric

semiconductor would allow us to polarize the detector. Detection of Xe scintillation (a phenomenon that only happens when a Xe decay takes place) allows triggering the start-of-the-event, which in turn allows us to change the sample polarization from positive (steady state) to negative (active state). The use of positive polarity between events repels positive ions and thus, prevent uncontrollable chelation of the FBI sensor.

ACTION TAKEN: To avoid any misunderstanding, we have removed the word “only” form the sentence (line 185, page 5).

2. *On page 9 the authors write that the observed chemical shift of the O1s core level induced by the deposition of BaCl₂ and NaCl is not a “just a doping shift” since there is a formation of a new O1s component in the XPS spectra. I think they should clarify what they intend as a doping shift. If they intend that it’s a shift induced by the presence of metal ion that produce a local dipole field that could be still the case depending from the distribution on the surface of the ions. Alkaline metals on surface are known to produce shifts of core levels of neighbouring atoms, effect that has been interpreted as a local work function modification. Furthermore, if the molecules containing Ba ions coalesce in islands, we could still have 2 peaks in XPS.*

It is well know that alkali metals on a surface (both adsorbed or intercalated) can induce surface state energy shifts, and therefore, can change the material work function. This effect is known as “doping”. For molecules on surfaces it is well stablished that, in photoemission, this variation of the work function is detected as a rigid shift of the molecular core levels (ACS Nano 7, 6914 - 6920 (2013)). The core levels, although shifted, on the other hand maintain the FWHM and shape. Thus, in the case of Ba (Na) interacting with FBI, in case of doping we may have a rigid shift of the O 1s, N 1s and C 1s toward higher B.E. of a similar magnitude for all of them, while maintaining in all core levels the shape and FWHM. On the contrary, we have observed different shift values for C 1s and N 1s peaks, as well as a broadening of the O 1s core level. As we discuss in the manuscript, this broadening and apparent shift of the core level is due to the growth of a new component at higher B.E. (chelated molecules) while the original component (non chelated) only decreases in intensity.

ACTION TAKEN: We have included the following sentence in the last line of page 10 (line 454): “...we determine that this shift is not just a doping shift, i.e. it is not just a rigid shift of all molecular core levels, but a chemical change of the molecules because a new component grows at a higher BE position.”

3. *On page 9-10 the concept that the O1s line width increases because of the formation of a second component is repeated twice.*

ACTION TAKEN: In lines 464-470, we changed the sentences “Once BaCl₂ is added to the sample, in the O 1s core level a second component grows at higher binding energy, which induces an increase of the peak width. At the latest stage of barium addition, the FWHM increases by 9% with respect to the initial state, which indicates that an extra component appears.” by “Once BaCl₂ is added to the sample, a second component grows at higher binding energy in the O 1s core level. This can be seen as an increase in the peak width of 9% FWHM with respect to the initial state.”

4. *On page 12 the authors claim that in this specific system they can detect conformational changes in STM images collected in constant current mode since there is “no variation in the electronic structure”. Is not very clear to me what they mean since, clearly, there are modifications of the electronic structure that, in fact, cause the change in fluorescence and the band gap, maybe they mean that there is no variation in the electron density of states detected by the STM in the specific energy range probed in those measurements.*

We thank the referee for this comment because, in fact, this is exactly what we want to say. We have rephrased this paragraph and modified the STM figure 4 to make this point clear enough. Moreover, since one of the image that it was shown in Figure 4 was acquired at a bias voltage above HOMO

orbital (-1.8V), we have decided to modify the figure and replace the Figure 4d by another chelated molecule measured at -0.5 V. Let us mention that both images for -1.8 and -0.5V were measured on exactly the same molecule.

With these actions each of the STM images presented in Figure 4 were acquired applying a tip bias voltages value inside the molecular band gap and far from HOMO or LUMO energy values, the up and down contrast in the images can be associated to the topography of the molecule and not associated to tunnel to/from different electronic states.

ACTIONS TAKEN: 1) We have changed the Figure 4d by one measured for $U = -0.5V$.

2) We have included in the caption of Figure 4 the bias and current used for acquiring each of the STM images.

3) On page 13, line 579, we have rephrased the sentence “since the images were measured at bias voltages well inside the molecular gap, i.e., with no variation in the electronic structure.” by “since the images were measured at bias voltages well inside the molecular gap, i.e., the image contrasts can be associated to topographic variations and not associated to tunneling processes to/from different electronic states.”

5. *On page 14 they write “The absence of substantial frontier molecular orbital gap upon complexation with Na^+ ”. It is not very clear to me what they mean, maybe the absence of modifications in the frontier orbital?*

The referee was right. We mean that there is no significant modifications of the frontier orbitals.

ACTION TAKEN: On page 14 in line 643, we added “modifications in the” frontier orbital.

6. *On page 16: the authors say that they “enhance” the contributions of the molecule to the O1s peak by subtracting the substrate contribution. I don’t think enhance is the write word, maybe highlight or select?*

ACTION TAKEN: in line 755, we replaced “enhance” by “highlight”.

Reviewer #2 (Remarks to the Author):

7. *Does the work support the conclusions and claims, or is additional evidence needed? I think the evidence presented is reasonable for the conclusions drawn. Although, I think the curves in Fig 4-e are mislabeled/mis-described in the caption. The curve with the HOMO-LUMO gap of 3.63eV (green) is the FBI-Ba2+ right ? The shift from 3.17eV for just FBI to 3.63eV for FBI-Ba2+ is the key result consistent with the blue shift of fluorescence observed for earlier studies in solution (Table 1). So please check/correct the caption and coloring of Fig 4.*

We thank the reviewer for pointing out this error. We have revised the figure and used the correct color for the spectra.

ACTION TAKEN A new version of Figure 4 has been included, with a corrected coloring of the spectra. We left the caption untouched

8. *This comment is beyond the scope of the paper which aims to demonstrate the FBI-Ba2+ fluorescence shift in UHV, and so should not be interpreted as a criticism of this paper but it would important in later studies to investigate if other species could result in a shift close to the Ba2+ shift, this paper shows that FBI-Na+ results in a different fluorescence shift. What about, for example, the daughter of 110mAg decay?*

As we have mentioned in the answer to referee #1, the mass spectra of the residual NEXT vessel gas do not show traces of metals that could chelate the crown-ether moiety. Moreover, the NEXT detector has not Ag elements that could produce Ag impurities.

On the other hand, NEXT selects events, which have exactly 2.5 MeV total energy and exactly two topologically connected electrons with no floating energy. Only two types of isotopes produce such events, all of the others being fully eliminated by the standard cuts in NEXT (no Barium tagging needed). The events are ^{214}Bi decays, which produce an energetic gamma (which later produces a single electron in the chamber) and ^{208}Tl (same process).

A decay of ^{110}mAg would not pass the cuts of energy and topology. To insist, Ba^{2+} must be detected in a time window compatible with a Xe-136 decay (in short: in coincidence with the two electrons although delayed coincidence). ^{110}mAg or any other isotope that we know of does not produce exactly two electrons of exactly 2.5 MeV (+- 20 keV) energy.

9. *Are there any flaws in the data analysis, interpretation and conclusions? Do these prohibit publication or require revision? Just my earlier comment, which I think is just a mislabeling in the caption of Fig 4. The feature associated with the HOMO level of the red curve of Fig 4e is a little difficult to see. Perhaps this could be better emphasized in the figure*

ACTION TAKEN: A new version of Figure 4 has been included, with a corrected coloring of the spectra. We have also highlighted the HOMO-LUMO positions by including new coloring shadow areas.

10. *Line 64 (abstract) suggest to spell out FBI*

ACTION TAKEN: This acronym was spelt out in lines 57-58. We changed the casing there.

11. *Line 87, ^{130}Te is the main isotope under investigation in the Te experiments suggest to mention that instead of ^{128}Te , also suggest to reference the Te experiments (CUORE and SNO+)*

We thank the referee for spotting this typo, indeed ^{130}Te is the main isotope for CUORE and SNO+.

ACTION TAKEN: ^{128}Te has been replaced by ^{130}Te on page 1 line 87. We have also included references to CUORE and SNO+ experiments (new Refs 15 and 16).

12. *Line 185, since only Ba^{++} and Na^{+} were investigated, I feel the statement "We demonstrate that only Ba^{2+} ions induce" is too strong. Suggest "Of the species investigated only Ba^{2+} ..."*

We redirect the reviewer to our answer to the first question of referee #1. He/she points out the same aspect.

ACTION TAKEN: To avoid any misunderstanding, we have removed the word "only" from the sentence (line 185 page 5).

13. *Line 229 typo structural  structurally*

ACTION TAKEN: in line 229, we changed "structural" by "structurally"

14. *Line 365 suggest to spell out binding energy (BE) apologies if I missed it earlier observation: Figure 2 was difficult for me to read on the printed version but fine in the electronic version*

BE was spelt out in line 266 (page 6).

15. *Line 439: I think I know what you mean, but I felt "a real chemical change" is rather loosely defined, suggest to rework the message of that sentence.*

This comment is in line with question 2 from the first referee. We rewrote this part to make it more comprehensive.

ACTION TAKEN: We have included the following sentence in the last line of 10 (line 454): "...we determine that this shift is not just a doping shift, i.e. it is not just a rigid shift of all molecular core levels, but a chemical change of the molecules, because a new component grows at a higher BE position."

16. *Table 1: Suggest to repeat in the caption the reference(s) for the absorption/emission measurements in solution results listed.*

ACTION TAKEN: we included the reference [5] here.

17. *Line 633 typo on substrate*

We do not see any typo in the whole paragraph for “substrate”.

18. *Fig. 5 caption got a little garbled in the pdf rendering I received*

ACTION TAKEN: we have checked to properly compile the caption.

19. *Line 671 suggest to define ϕ (it is defined in methods but could be helpful to define it here or forward reference to methods)*

ACTION TAKEN: we have added the definition in line 410-423 and a reference to methods in line 502.

20. *Suggestion: If available, some comment in the conclusions about the reproducibility could be helpful, for example how many samples were tested? Were possible aging effects investigated?*

In addition to the experiments included in the manuscript, many control experiments were done. Let us mention that in a standard surface science experiment the same sample, cleaned and refreshed to expose an ideal surface, is used several times. Evaporation calibration was done, independently of the substrate by quartz microbalance and reproducibility was proven by quartz microbalance and XPS peak intensity. The samples are not replaced after sublimation, instead they are cleaned. In total more than ten full experiments were performed for Cu (7 chelation tests), Au(111) (3 chelation tests) and ITO (1 chelation test). Moreover, chelation on polycrystalline Au surface, TiO₂ and quartz was also tested as well as the control experiments evaporating just BaCl₂ (NaCl) directly on the bare substrates. Regarding aging, the XPS measurements to characterise the chelation took about 12 h, so ageing processes and radiation damage we discarded up to that period. The evaporations and measurements in figure 2d took a total of 4 days of XPS acquisition (free state + 6 progressive BaCl₂ doses, each one of about 12 h). However, no experiments were performed to address this question specifically.

ACTION TAKEN: On page 17, line 815 the following comment has been included: “The agreement between experiments performed on different substrates and subsequent preparations confirmed the reproducibility of the evaporation and chelation processes.”

Reviewer #3 (Remarks to the Author):

21. *Regarding molecular sublimation: XPS intensity ratios and high-resolution STM images of individual FBI molecules are used to confirm intact adsorption of the molecules on Au. However, matching XPS intensity ratios might not exclude fragmentation of the molecules on the support. The STM images show the coexistence of individual molecules and molecular aggregates. Could the structure of the molecular islands be resolved and can it be excluded they are formed by molecular fragments? The STM analysis in Fig. 3 and Fig. 4 addresses subtle details of the appearance of the molecules and relates these to the molecular structure. In the molecular islands, quite an inhomogeneous contrast is observed. This raises the questions if the information extracted from selected single molecules is representative (see additional comment below).*

While it is true that it is almost impossible to exclude any fraction of molecular fragmentation, the experiments do provide no clear evidences that this is taking place. First, because the evaporation process is very stable and very reproducible (same rate for same sublimation temperature). This rules out fragmentation inside the evaporation cell because if, during the evaporation process the molecules get broken, then the evaporation fragments must sublime at different sublimation temperatures, and the evaporation would become non-reproducible. The FBI related compounds we used as fluorophore reference at manuscript's Fig 3 sublime at 60-80°C, while the intact FBI molecule sublimates at 140-150°C.

Figure 2: Detailed Zoom images of molecules at islands. The false colors were used as eye guide to identify each molecule

Figure 3 STS spectra measured on the isolated molecules (upper inserted image) and on top of the molecules at the molecular island edge.

Figure 4: STM images of the two FBI derivatives where the tendency to form dimmers or tetramers is

Moreover, fragmentation should have an effect in the XPS spectra: the core level intensities should become evaporation-dependent and, thus, the stoichiometry would change for subsequent experiments. However, this is not the case. The measured XPS ratios between molecular elements (C, N and O) as well as their energy position and shape, are perfectly reproducible for consecutive evaporations. Thus, if any molecular fragmentation took place, it would do so only induced by the surface. It is at this point, where comparison with STM is important.

As the referee mentioned, the STM images show individual molecules coexisting with islands of molecular aggregates. At these islands, due to the lack of molecular order, identification of individual molecules is complicated and subject to the beholder eyes interpretation. As an example, at Figure 2 we have included three zoom images of some of the molecular aggregates visible in Fig. 3 of the manuscript. In these zooms, we have identified, by means of artificial colors, the features which resemble the individual molecules. Of course, this is an interpretation, but what it is clear is that it is always possible to identify triangular features; some of them appearing lower and flatter than the others, in good agreement with individual molecules.

Moreover, the STS taken on molecules inside the islands perfectly compare to the spectrum measured on the isolated molecules, which confirms that they are not broken (Figure 3).

The last evidence to exclude fragmentation comes from the comparison of low magnification STM images for the FBI with those obtained for the only-fluorophore-FBI molecules specifically synthesized without the aza-crown ether component. Figure 4 shows two STM images of the evaporation of these molecules from Fig. 3 c-d of the manuscript. Both only-fluorophore-FBI molecules tend to form dimers or tetramers instead of islands.

ACTIONS TAKEN:

1) We have added extra STM and STS images as supplementary information Fig S3 and S4 (the three figures included in this answer).

2) We have included the following sentences in the main text:

- Page 7, line 294 “Identification of individual molecules on the islands, is complicated due to the lack of molecular order, and it is subject to the beholder eyes interpretation. However, it is possible to identify the features which resemble the aspect of the individual molecules (See figure S1 in Supp. Info). With this we rule out molecular fragmentation at the surface.”
- Page 8 line 337 “It is remarkable that none of these FBI derivatives forms molecular islands. Instead, only dimers or molecular tetrameters are distinguished in the STM images (see Supp. Info.). This supports the previous assessment, that FBI molecules remain intact (non fragmented) after sublimation”
- Page 19 line 845 “FBI evaporation was very stable and very reproducible, since the same rate for the same sublimation temperature was monitored. This rules out fragmentation inside the evaporation cell because if, during the evaporation process the molecules get broken, then the evaporation fragments would sublime at different sublimation temperatures, and the evaporation would become non-reproducible The simpler FBI-derivative molecules sublime at 60-80°C, while the intact FBI molecule sublimates at 140-150°C.”

22. *Regarding chemical demonstration of chelation: The O 1s peak shift is used as fingerprint for the chelation of the metal ion. Why are no XPS data of the metal ions included? These could provide valuable information on the chelation, as discussed for example in ref. 36 for the Na 2p signature.*

Following the reviewer suggestion, we have decided to include XPS spectra measured for Ba 3d_{5/2} core level as Fig. S2 of the Supp. Info. Although we measured Na 1s as well, we did not collect high-resolution experiments in all the cases. We thank the referee for this suggestion. We did not consider including them in the original version because they do not show relevant shifts when increasing the dosing. However, the referee is right that it could provide valuable information to confirm that the oxidation state of the metal ions remains as 2+ (the desirable one in the context of NEXT purposes).

Figure 5: O 1s and Ba 3d_{5/2} core levels measured for increasing dosage of BaCl₂ on 0.9 ML of FBI on Au(111). The reference of BaCl₂ deposited directly on Au(111) was also included as reference.

Figure 5 of this response shows the Ba 3d_{5/2} core level evolution for incremental amounts of BaCl₂ doses ions from 0 to almost 0.5 Ba²⁺ ions per FBI molecule. These spectra were collected together with those of O 1s core level shown in Figure 2d. As can be seen, the Ba 3d core level position does not vary when increasing the dose over the FBI layer. Compared to the peak from BaCl₂ deposited on bare Au(111) (red control spectrum), there is only a slight shift of 0.1 eV toward lower BE. The position unchanged and the good agreement with reference BaCl₂ clearly demonstrate that the oxidation state remains constant during experiments.

ACTIONS TAKEN:

- 1) We included the evolution of the Ba 3d_{5/2} spectra as a function of the BaCl₂ deposition in the Supplementary Information fig S2.
- 2) We have included the following sentence in the main manuscript (page 11, line 484): “During the incremental deposition of BaCl₂, the Ba 3d core level was also controlled. Here only the intensity increases, but there is no significant change in the core level position. This confirms that Ba maintains the 2+ oxidation state during the whole process. The evolution of the Ba 3d_{5/2} core level is included in fig. S2 of the Supp. Info.”

23. *No XPS data for complete chelation ($\phi=1$) are shown for the Au support. In Fig. 2e ϕ reaches only a value of 0.42. A data set with larger ϕ might be instructive. If selective chelation is achieved, one might expect a modification in the XPS data for $\phi>1$.*

The referee’s observation is correct. In the manuscript discussion, for the case of Au(111) we just showed data for BaCl₂ doses lower than 1. The reason was we tried to emphasize the initial stages of chelation. However, following the referee’s comment, we have slightly modified the discussion in the revised version and we have included O 1s spectra measured for doses above 1 in the Supp. Info. (Fig S1). In particular we include the spectrum for $\theta_{Ba} = 3$ (Figure 6 of the present response). The decomposition of the peak demonstrates that the component related to chelated species grows. However, in any case the un-chelated component disappears completely. This reveals that the

chelation efficiency in our measurement conditions is not 100%. For the fitting, the green component is always fixed in position (532.9 eV) and width (1.07 eV). The blue and red components are free and they are both centered on 533.9 eV.

Figure 6: O 1s core level evolution for increasing dose of BaCl₂.

ACTIONS TAKEN:

- 1) We have included a new figure in Supplementary Information, Fig S1.
- 2) We have included the following sentence in the main manuscript (page 11, line 473): “The position of this new component remains constant even for θ_{Ba} doses above one Ba²⁺ ion per FBI molecule. This is shown in Figure S1 of the Supp. Info. There, the O 1s core level is compared for $\theta_{\text{Ba}} = 0.8$ and $\theta_{\text{Ba}} = 3$ on 0.6 ML of FBI on Au(111).”

24. *When deconvoluting the O 1s signal, the authors assign a component at around 533.9 eV to the crown ether chelation, referring to ref. 36 that addresses Na chelation. The respective binding energy in ref. 36 however is different, as is the ion. The authors might clarify if the peak position of 533.9 eV was obtained as a free parameter in the fitting procedure or if it was fixed based on some considerations.*

The peak position of 533.9 eV was obtained as a free parameter. Only the other component (at 532.95 eV) was fixed for the chelated molecule as mentioned in the text. On reference 36, the maximum of the O 1s peak is indeed closer to 533.5 eV (for maximum doping with Na), and not on 533.9 eV. When comparing to Ref. 36, we have to keep in mind that we are working with different molecules. In our case, the crown-ether group is covalently linked to a fluorophore, while in the case of Ref. 36 independent crown ether groups were co-evaporated with porphyrins. It is worth remembering that in Ref. 36 crown ether groups were oriented parallel to the Au substrate without any possibility to move or rotate, while in our case the crown ethers have more flexibility and mobility, and can adopt any conformation with respect to the surface. Therefore, the important point here is that the trend toward higher binding energy is the same as in our case.

ACTION TAKEN: We have added “These values were obtained from the fit as free parameters.” in line 992, page 21.

25. *In the conclusion, one reads about the important capability of the aza-crown ethers to interact with "many different ions". However, the manuscript only addresses two ions, so one might consider to adapt the wording.*

We agree with the referee. This sentence is misleading. We have reworded it, clarifying this expression. From previous works of chelation in solution, we know that all the alkali metals chelate the molecules (although not all of them induce fluorescence shifts). On surfaces, We have also tested chelation using Fe^{++} (by sublimating FeCl_2) in the context of investigating an alternative on-surface coordination reaction. By combination of XPS, UPS and NEXAFS we can confirm that chelation also takes place with this ion (work in preparation).

ACTION TAKEN: We have rephrased the last paragraph of the conclusions (line 793, page 18). The text now reads "In addition to the tests presented here with Ba^{2+} and Na^+ ions, the aza-crown ethers group has demonstrated to have a high affinity to chelate alkali ions in solution and even transition metal ions (publication in preparation) . This high affinity for trapping different ions could have important applications in drug carriers \cite{uchegbu_non-ionic_1998} or photo-switching devices \cite{malval_photoswitching_2002, uda_membrane_2005}"

26. *Regarding molecular structural rearrangement: Based on the STM data provided in Fig. 4, the authors discuss the conformations of the three configurations. This seems a bit speculative. Distinct differences are discernible between all three images. STM image simulations based on the proposed conformations would be clearly beneficial to support the conclusions. Furthermore, Fig. 3 shows individual molecules as well as molecular islands for FBI only, indicating different conformations occur even for chemically identical species. How characteristic are the individual STM images shown in Fig. 4 for the respective species?*

This question is very much related to the questions 1 and 4 from referee. 1. We thanks these comments since they help us to improve the comprehension of STM data, especially with the suggestion to compare our experimental STM images with STM simulations.

We agree with the reviewer that the interpretation of topological STM images can be in many cases speculative. In this case, as he/she points out, many possible molecular adsorption positions, molecular orientations and intramolecular conformations can influence the STM images and make interpretation ambiguous. However, even in this scenario, there are some very reproducible features: two close triangular-shaped regions, one being darker than the other. This is observed for the isolated molecules as well as in the dimers or in islands (see Figure 2 of present response).

Regarding STM images simulations, in this particular case where a zoo of possible molecular-substrate configuration should be considered, these calculations are almost unaffordable. What we have simulated are the molecular conformation in gas phase. Figure 7 shows the MM–MC conformational analysis (OPLS3e force field) of unchelated and chelated FBI molecules. The Ball & Stick representation corresponds to the most stable conformation. For FBI molecule, there are thinner sticks that corresponds to ten other most stable conformers within 0.0-4.8 kJ/mol. Electron density surface associated with the most stable conformer is also shown and can be compared with the aspect of the STM images shown in Fig.4 of the manuscript. As it can be seen, the aspect of the molecules is not very much different in the fluorophore region (illustrated with the superposition of the FBI profile as yellow edge in the three species), while they are slightly different in the crown-ether region. These subtle differences are similar to the one we have observed in the experiments and are the reason why we have used the spectroscopy for the final molecular assignment.

Figure 7: Most stable molecular conformations, calculated in gas phase for chelated and unchelated FBI molecules. Top and bottom images represent the ball & stick illustration. In the middle the electron density molecular states are presented.

ACTIONS TAKEN:

- 1) We have included a discussion of the bound states calculations in Fig. S5 of the Supporting Information.
- 2) We have modified Figure 4a) to include the most stable calculated structured instead the schematic representation of the molecular composition, as it was shown before. Figure caption was modified in accordance with changes to the figure.
- 3) We have included the following sentences in the main text:
 - Page 12, line 536: “Figure 4a shows the schematic representation of the most stable conformation calculated for the molecules in gas phase. In all cases, the crown ether ring is out of the plane of the fluorophore, and in the case of the Ba²⁺-chelated molecule, this group is folded over the fluorophore. The details of the distances extracted from the calculations are included in fig. S5 of the Supp. Info. Thus, for molecules lying on a surface, we expect to have the fluorophore parallel to the substrate, and the crown-ether region out of the plane, as we see in the STM images, Figure 4 b-d.”
 - Page 12, line 608: It is worth mentioning that the aspect of these STM images resembles the electron density simulations associated with the most stable conformers shown in Figure 4a (see also fig. S5 of the Supp. Info.). Even with this evidence, unambiguous association of chelation and molecular rearrangement can only be done by comparing the spectroscopy scanning tunneling spectroscopy (STS).

27. *The C 1s peak shapes (Fig. 2) seem very similar for all three configurations. Would one not expect distinct differences between "flat" and "more three-dimensional" conformations, with C atoms lifted from the metallic support in the latter case?*

Yes, the referee is right. We do expect those differences, and there is indeed a small shift in the C 1s peak of about 0.2 eV. In the text we did not mention this aspect in much detail because, due to our low resolution, it was difficult to claim an incontestable evidence. When we fit the C 1s core level we use two components in both the free and chelated molecules, as can be seen in the Figure 8. For the free FBI, we find a component at 284.5 eV and the other at 285.9 eV. For the Ba²⁺-chelated molecules we fix the component at 284.5 eV and find another at 286.1 eV as a free parameter. The relative intensity of the latter component also increases with the Ba²⁺ dose. This evidence could be interpreted as C atoms far from the surfaces even for the unchelated molecules. Considering this interpretation and our limited energy resolution of 0.05 eV, we decided not to highlight this result and move it to supplementary information.

Figure 8 C 1s core level spectra measured for unchelated FBI molecules and for increasing doses of BaCl₂. The curve component fitting is included to highlight the increase of the component at higher B.E.

Just for the reviewer information, using other molecules of the FBI family (isomers of the molecule presented in the actual manuscript), we have recently carried out NEXAFS experiments and we have observed change in the orientation of the π -orbitals after capturing FeCl₂. This change can be also associated with changes in the C 1s and N 1s spectra, in agreement with the reviewer's comment. This data is still being processed.

ACTION TAKEN: On page 10, line 446, we have replaced the sentence “Figure 2b shows the O 1s, N 1s, and C 1s, whose shifts toward higher BE are distinguished, mainly on O 1s.” by “Figure 2b shows the O 1s, N 1s, and C 1s core levels. Although the three of them exhibit small differences compared to unchelated molecules, the shifts toward higher BE are better distinguished on O 1s.”

28. *Additional information about the STS spectra in Fig. 4 should be provided. Do these spectra reflect a spatial average over the molecule or are they characteristic for specific positions?*

All the spectra were measured on the fluorophore region, because in the crown-ether region reproducibility was not achieved due to its high structural flexibility.

ACTION TAKEN: We have included the term “in the fluorophore region” in the sentence “Figure 4e shows the associated STS spectroscopy measured for the three molecules in the fluorophore region” line 638, page 14

29. *On page 12 one reads that the STM images are comparable, as they "were measured at bias voltages well inside the molecular gap". Is this really true? For example, the caption of Fig. 4d gives a bias voltage of -1.8 V. Judging from the corresponding spectrum in Fig. 4e, this is outside the molecular gap. This would affect the comparability of the apparent heights.*

This question was also pointed out by reviewer 1 (question 4). The referee is right: in the original version, the Figure 4d was measured for a bias voltage above the molecular gap, and this could induce to errors. In the revised version we have replaced the image 4d by a new one, measured at bias voltage -0.5 V, i.e. in the molecular gap.

ACTION TAKEN: We have corrected figure 4d) and the caption correspondingly.

30. *On page 14, one reads that "the measured HOMO-LUMO gap values are in agreement with the $\lambda_{\max}^{\text{emi}}$ spectra measured in solution", the conclusions claim they are "perfectly consistent". How do the authors reach this conclusion? Both the absolute values (for example 391.1 versus 489) and the trends (STS: 3.17 and 3.31: different, emission 489 and 489: identical) do not really match. Why do the authors expect an agreement at all? The different spectroscopic methods and the environment should impact the experimentally measured gaps.*

The way this sentence was presented in the original manuscript induced to errors in the interpretation. The referee is right that it is not possible to directly compare the fluorescence experiments with the STS spectra because the physical origin of the processes are different, and of course the different environment can impact the gaps. What we wanted to highlight with the comparison was that the trends observed in both methods matches, and that blue shifts in the fluorescence are somehow also reflected in band gap increments in the STS.

ACTION TAKEN: We have rephrases caption of Table 1 as well as the sentence "The measured HOMO-LUMO gap values are in agreement with the..." (line 655, page 15) has been changed by "Thus, the measured HOMO-LUMO gap values show a blue shift tendency for chelated molecules, mainly for λ_{app} -chelated FBI, which follows the trends observed in fluorescence spectroscopy for the same molecules in solution. The absolute values from both spectroscopic methods can not be directly compared because they are based in different physical processes. Nevertheless, the trends measured by both methods are comparable. In order to illustrate this similarities, Table 1 summarizes the absorption and fluorescence emission maxima $\lambda_{\max}^{\text{emi}}$ and $\lambda_{\max}^{\text{abs}}$ associated to the transition of electrons from HOMO to the LUMO+n states (absorption), to LUMO, to the highest HOMO transition available by geometrical restrictions (fluorescence)."

31. *Regarding chelation on different surface supports: The authors compare different supports (Au, Cu, ITO), concluding based on O 1s XPS data that chelation is "independent from the choice of the substrate". However, the Ba coverages are strikingly different for the substrates (for example $\phi=0.8$ on Au versus $\phi=17.5$ on ITO). How is such a comparison justified? Can it be excluded that "overdosing" ($\phi>1$) modifies the O 1s spectra?*

The referee is right. This question is related to a previous one (question 23), where the referee asked about how the core level shifts for low and high BaCl_2 dose compare with each other. As we showed the shifts are the same for Au(111). For the case of ITO we did not perform an exhaustive experiment comparing O 1s evolution for very low BaCl_2 doses because of experimental restrictions: ITO has a strong O 1s peak related to the substrate. Therefore, in order to appreciate the differences, we chose to go for high doses.

In the case of Cu (111) we do have data for lower doses of Ba^{++} , see Figure 9, where the shift is visible. However, the detection of undesirable Cu contamination in the O 1s core levels would difficult the core level shift demonstration. This is an unavoidable effect associated to the experiment. To avoid cross talking of the evaporators, the BaCl_2 is located in a chamber module different to the analysis module. This implies transfer of the sample from one to another which produces a small, but detectable, Cu oxidation. This contamination yielded the shoulder at low BE (531 eV).

Figure 9: O 1s and Ba 3d_{5/2} core levels measured for increasing dose of BaCl₂ on 0.9 ML of FBI on Cu(111).

ACTION TAKEN: We have included a new figure for the O 1s evolution on Cu(111) in the Supp. Info. (fig S6). In addition, the following sentence was included in the main text (line 724, page16) “Although here we only discuss the results for high θ_{Ba} , a similar shift evolution as the one observed for low doses on Au(111) (Fig. 2d) was measured on Cu(111) (see fig. S6 of the Supp. Info.)”

32. *In the conclusion one reads that “..conformational changes occur upon chelation, independently of the substrate..”. Is this conclusion justified? The XPS data alone (Fig.5) do not seem to address the conformation*

The referee is right. The conformational changes were only observed for Au(111), but we have no experimental evidences of them on Cu(111) and ITO, where we could only probe the chemical changes. For this reason, we have decided to rewrite these conclusions.

ACTIONS TAKEN:

- 1) Line 772, page 17, we have replaced “Chemical and conformational changes occur upon chelation, independently of the substrate where molecules were deposited” by “Here, by the combination of highly sensitive surface science techniques, we have unraveled the chemical and conformational changes that occur upon chelation of FBI molecules. We started with Au(111), a well-known surface that served as model, and moved on toward more suitable surfaces.”
- 2) Line 781, page 17, we have modified the sentence “ Moreover, the measured variation in the molecular HOMO-LUMO gap is perfectly consistent with the observed bicolour property of the sensor for Ba²⁺ ions compared to the absence of shift observed and predicted for other ions, such as Na⁺.” by “Regarding the bicolour property of the sensor: fluorescence emission could not directly measured because of the metallic character of the substrates. Nonetheless, the measured variations in the molecular HOMO-LUMO gaps are perfectly consistent with the observed emission shift for Ba²⁺. Moreover, they are also consistent with the absence of such shift, observed and predicted for other ions, such as Na⁺.”

Additional (minor) comments:

33. *The role of the Cl, provided by both the BaCl₂ and the NaCl, should be discussed. Cl is described as “passive spectators”. Is this assumption justified on a surface? In the method section, one reads that NaCl was used, while the text mentions sublimation of Na⁺. This should be clarified.*

Starting by the last comment, the reference to sublimation of Na^+ was a typo, and has been corrected in the text.

Regarding the role of Cl, in our previous publication, Ref. 5, the role of Cl in the chelation process was computed, (and included in the Extended Data Figure 3 of Ref 5). Simulations in gas phase determined that the Cl^- anions are playing no role in the processes. Experimentally, we have not addressed whether the presence or absence of the chlorine modified the chelation. However, what we can say from the XPS experiment data is that Cl_2 are somehow “passive spectators”. Moreover, in many cases, once chelation takes place, these Cl atoms are detached from the molecules and even desorbed. We extract this conclusion from the fact that the Cl:Ba ratio changes when BaCl_2 is deposited on FBI-functionalized surfaces. While sublimation on bare substrates kept the expected 2:1 ratio, there is a decrease of the Cl amount when sublimated on FBI. In Figure 10, it can be seen how the Cl amount is lower, compared to Ba, when evaporation is done on 0.9ML of FBI on Cu(111). These results are very reproducible.

Figure 10: Comparison of Ba 3d and Cl 2p core levels for BaCl_2 deposited on bare Cu(111) or on 0.9 ML of FBI on Cu(111).

Apart from these experiments, as mentioned previously, we have carried out further chelation tests using other molecules of the FBI family using FeCl_2 in a synchrotron. We have clearly observed dechlorination induced by FBI. We are not going to enter into the details for those experiments, but just to highlight to the referee the aspect we consider more relevant related to this point. First, we observed a partial Cl desorption (reflected in the non-stoichiometric Fe:Cl ratio). Second, we clearly resolved a dehalogenation process (the remaining Cl atoms are detached from the Fe and then shifted in the XPS Cl 2p core level toward the Cl-Au position). Last and most importantly, post-annealing processes yielded completely different results for FeCl_2 on Au(111) compared to FBI/Au(111). Heating the sample over 70-80 °C produces a sheer decrease in the intensity of the Cl 2p peak (while Fe remain intact) for FeCl_2 deposited on FBI/Au(111) surface. On the contrary, when FeCl_2 is deposited directly on Au (111), the intensity of the Cl 2p peak remains constant for temperatures over 120 °C, indicating that FeCl_2 remains intact on the surface. Figure 11 illustrates these different behaviors: the Temperature dependent XPS spectra is shown, measured for the Cl 2p core level (X axis corresponds to BE around the Cl 2p core level, y axis is the acquisition temperature and the core level intensity is represented with the color scale). We are still processing this data, but it points toward confirmation that Cl is not playing any role in the chelation process.

Figure 11: Temperature-dependent Cl 2p core level measured after FeCl₂ deposition on Fbi/Au(111) substrate and on Au(111).

ACTION TAKEN: We have consired that it was not necessary to take any special action regarding this point, because the role of Cl (as well as the role of Xe) were discussed in our previous publication, Also, because a detailed study of chelation with FeCl₂ is in progress and it will address this topic.

34. *On page 16, the residual contamination of Cu by O reflected in a Cu₂O contribution is mentioned. Is this related to the dosing procedure or does this reflect contamination issues with the bare Cu sample?*

The bare Cu sample was clean (see for example Figure 9 of the present document), and it did not present the peak associated with the contamination. The contamination appeared after evaporation of BaCl₂, which took place in a separate chamber module than that of the characterization chamber.

35. *Fig 4 is confusing. The three columns include three situations each. While the left column and the middle column match, the spectra in the right column are not consistent.*

ACTION TAKEN: Figure 4 has been remade in order to address all the comments mentioned by the reviewers.

36. *In surface science, "phi" usually describes the work function. Here, it is used for coverage. One might consider adapting the notation.*

We appreciate this comment and we have proceed by changing Φ by θ , as in ref [36].

ACTION TAKEN: Now we use the notation θ_{Ba} and θ_{Na}

37. *The manuscript should be carefully proofread, there are several typos, formal issues (e.g., caption Fig. 5), and a figure panel not mentioned in the text (Fig. 2e).*

We have revised the text and we hope now we have corrected the typos and formal issues pointed by the reviewer.

REVIEWER COMMENTS

Reviewer #2 (Remarks to the Author):

I'm happy with the changes made by the authors. I thank them for their efforts to address my comments and for summarizing them clearly in their response letter. I recommend publishing the paper.

Reviewer #3 (Remarks to the Author):

The authors provided a comprehensive point by point reply, resolving some of the issues raised in the report (such as the intact deposition of the FBI). Unfortunately, the additional information provided by the authors did not help to convincingly support all conclusions. Thus, the reviewer can not recommend acceptance of this manuscript in the present form.

Some issues the referee considers not fully resolved are listed below, referring to the numbers used by the authors in the point by point reply.

23. & 31. A key claim of the manuscript is the successful "chelation of FBI indicator by Ba²⁺ ions" on different supports. As a proof of chelation, the O 1s XPS spectra are used and shifts are taken as "fingerprint of the chelation". The additional data (Fig. S1) show that the O 1s component assigned to chelation increases even well above the "saturation level" ($\Theta_{\text{Ba}} \geq 1$). The reply states "...the component related to chelated species grows." Concomitantly, a contribution of native FBI is observed well above $\Theta_{\text{Ba}} = 1$. The authors write "...a maximum of 3 Ba²⁺ per molecule are needed to saturate the chelating vacancies." Such a capture efficiency well below 100 % indicates that for every Ba ion chelated to the FBI, several additional Ba atoms or BaCl₂ molecules occur on the surface. This is not really emphasized in the manuscript. It also means that no data on fully Ba-chelated FBI on Au is provided. How can it be excluded that excess Ba, extensively available on the surface, affects the O 1s signature? As exclusively submonolayer FBI coverages are applied, Ba might aggregate at the periphery of aza-crown ether groups, potentially contributing to the modification of the O 1s signature. In such a situation, the O 1s spectra would not be a reliable fingerprint for chelation. Additional XPS data, covering samples with Ba-saturated FBI on Au, or STM images, might clarify this issue.

22. The authors now provide Ba 3d_{5/2} spectra and conclude that the Ba ion remains in a 2+ oxidation state upon chelation. The spectra only represent a Ba coverage up to $\Theta_{\text{Ba}} = 0.42$. Thus, potential modifications in the spectral signature at saturation (as for example demonstrated in ref. 38 for Na) remain elusive. Additionally, the Ba 3d_{5/2} binding energy seems to be rather insensitive to the oxidation state, with similar binding energies reported for metallic, elemental Ba (<https://srdata.nist.gov/xps/ElmSpectralSrch.aspx?selEnergy=PE>). Thus, the additional data do not provide a proof of selective chelation and the identification of a 2+ oxidation state seems not fully reliable.

26. & 29. The addition of the electron density contours is instructive. Nonetheless, the authors now indicate that an "unambiguous association of chelation and molecular rearrangement" can not be done by STM, but only by STS. Thus, additional STS data providing some statistics on several molecules and a refined analysis (e.g., addressing the uncertainty in the determination of the HOMO onset for FBI-Na⁺) would be important to substantiate the conclusions about chelation and molecular rearrangement. It also does not seem immediately clear why spectra recorded on the fluorophore region, which is claimed to be planar and interacting with the Au support in all three cases, should be representative for the structural rearrangements that mainly occur on the crown ether moiety (where also the apparent height is measured). Even though Figure 4 was corrected, the statement "... were measured well inside the molecular gap" still seems not really appropriate. Fig. 4e) (green spectrum)

suggests that the tail of the LUMO contributes at a bias voltage of 1.4 V (c).

30. The description of the comparison of STS and spectroscopy in solution was improved. Nonetheless, some of the statements still seem confusing and not fully supported by the data. Focusing on the trend of the gap, table 1 shows considerable blue shifts for Ba-chelated FBI (49.5 nm), but also for Na-chelated FBI (16.5 nm). The optical spectra mainly show a blue shift for Ba-chelated FBI (Emission Ba: 61 nm, Na: 0 nm). The conclusion reads "Nonetheless, the measured variations in the molecular HOMO - LUMO gap are perfectly consistent with the observed emission shift for Ba 2+. Moreover, they are also consistent with the absence of such shift, observed and predicted for other ions, such as Na+." While one might argue about the term "perfectly consistent" when discussing a trend, the second sentence seems to contradict the blue shift observed in the HOMO - LUMO gap for Na. And it also does not seem in line with the statement on page 15 "Thus, the measured HOMO - LUMO gap values show a blue shift tendency for the chelated molecules, mainly for Ba2+-chelated FBI, which follows the trends observed in fluorescence spectroscopy for the same molecules in solution". As the authors emphasize the important "bicolour property of the sensor" and as the discrimination of ions is crucial for the detection scheme, this issue is relevant.

Minor comments:

- It might be appropriate to mention the Ba²⁺ coverage used for the STM imaging in Fig. 4.
- The referee does not recognize the benefits of using a colored background in Fig. 2d/e and Fig. 4e. It makes it hard to read the labels (Fig. 2) and the additional colors (Fig. 4), not matching the colors of the spectra, are rather confusing.
- Certain wordings seem a bit strange. Line 312: "approximately resolved"; line 490 "core level was also controlled."
- The referee suggests to refrain from citing unpublished work in the conclusions.

Additional comment: Reviewers invest quite some efforts in assessing this manuscript, as a service for the authors. Accordingly, authors would be expected to carefully read their manuscript before submission, as a service for the referees. This does not seem to be the case for this manuscript, which includes several careless mistakes. Examples:

Line 299: S1 should read S3

Line 310: "On the right" isn't this "on the left"?

Line 314: "region is" or "regions are"

Fig. 2: inconsistent notation (Θ versus Φ)

Line 442: bracket and punctuation mark missing

Line 451: punctuation mark missing

Line 494: S1 should read S2

Line 626: "the spectroscopy scanning tunneling spectroscopy"

No reference to Fig. S4 in the main text

Caption Fig. S3: "Red squares show..", there are no red squares in the figure.

Caption Fig. S5: Does the bottom panel in b) really represent the electron density associated with the conformer show in a)? The labeling seems wrong.

We want to start by expressing our gratitude to the referee for his/her observations. His/her remarks made us think about a better way to explain the experimental observations, and also prompted us to consider new calculations and experiments. We think the referee's recommendations have improved the manuscript's thoroughness and clarity.

The authors provided a comprehensive point by point reply, resolving some of the issues raised in the report (such as the intact deposition of the FBI). Unfortunately, the additional information provided by the authors did not help to convincingly support all conclusions. Thus, the reviewer can not recommend acceptance of this manuscript in the present form. Some issues the referee considers not fully resolved are listed below, referring to the numbers used by the authors in the point by point reply.

1. *A key claim of the manuscript is the successful "chelation of FBI indicator by Ba²⁺ ions" on different supports. As a proof of chelation, the O 1s XPS spectra are used and shifts are taken as "fingerprint of the chelation". The additional data (Fig. S1) show that the O 1s component assigned to chelation increases even well above the "saturation level" ($\Theta_{Ba}=1$). The reply states "...the component related to chelated species grows." Concomitantly, a contribution of native FBI is observed well above $\Theta_{Ba}=1$. The authors write "...a maximum of 3 Ba²⁺ per molecule are needed to saturate the chelating vacancies." Such a capture efficiency well below 100 % indicates that for every Ba ion chelated to the FBI, several additional Ba atoms or BaCl₂ molecules occur on the surface. This is not really emphasized in the manuscript. It also means that no data on fully Ba-chelated FBI on Au is provided. How can it be excluded that excess Ba, extensively available on the surface, affects the O 1s signature? As exclusively submonolayer FBI coverages are applied, Ba might aggregate at the periphery of aza-crown ether groups, potentially contributing to the modification of the O 1s signature. In such a situation, the O 1s spectra would not be a reliable fingerprint for chelation. Additional XPS data, covering samples with Ba-saturated FBI on Au, or STM images, might clarify this issue.*

Referee suggested that the excess of BaCl₂ around the crown ether group could modify the O 1s position even when chelation was not occurring. This would be clearer in the submonolayer FBI samples where there is more space between molecular islands to have the BaCl₂. He/she asks for extra STM or XPS data to provide a more clear evidences to support our statement that O 1s core level shift is a XPS fingerprint of molecular chelation.

From the STM point of view, images does not help to clarify and/or exclude the lateral interaction as responsible of the chelation shift. As can be seen in the Figure 1 of the present answer, the STM images of the functionalized sample after BaCl₂ sublimation display considerably more individual FBI molecules than before and also smaller islands. The islands have a different aspect compared to before and could corresponds to BaCl₂ islands surrounded by molecules. Whether these molecules in contact with the BaCl₂ are chelated or not or present spectroscopic fingerprint comparable to chelated molecules (and related to the changes in the XPS spectroscopy) we cannot know. We could not obtain reproducible STS spectra for them because the tip was not stable, and, hence, we only to get spectra for individual chelated molecules (now extra STS included in the new version of Figure S5).

Mention here that for STM, in order to have enough clean Au surface to prepare the tip and because the objective is the characterization of individual molecules, we have never used FBI coverage higher than 0.4ML, while for XPS we have data for experiments close to the ML regime (0.9ML). Under these conditions, the amount of molecules that can be in contact with BaCl₂ islands is minimum. Therefore, this amount could not explain the variations observed in the XPS.

Figure 1: STM images of FBI submonolayer before (a) and after BaCl₂ sublimation (b). The arrows indicate the isolated FBI (chelated) molecules.

Regarding XPS data, referee asks for results on a fully covered FBI layer and/or after dosing $\theta_{\text{Ba}} \geq 1$. This is, in fact what Figure S6 shows: O 1s core level before and after chelation with nearly $\Theta=2$ BaCl₂ of 0.9ML of FBI (almost full FBI layers) on Cu(111). For the almost monolayer FBI layer with saturated Ba ions we still see the core level shift although. Situation for Au substrate was similar. For 0.9ML (as well as for 0.6ML), we observe the shift from very low doses up to $\theta_{\text{Ba}}=3$ (results now included in a revised version of Figure 2 in the manuscript). In all cases, the O 1s peak after chelation is broader and the fitting always results in two components: one at roughly 530 eV associated to non-chelated molecules, and the other at around 533.9 eV consistent with chelated crown ethers (Figure 2 of the manuscript).

In fact in any of the experiments, we have performed so far, the 100% chelation was observe. At this point, we should emphasize that, of course, there is room for improvement in chelation efficiency, including altering the anchoring of molecules on surfaces, changing their structures, or co-depositing with another organic layer as a model. However, the optimization of the chelation efficiency is out of the scope of the present manuscript.

Since experimental evidences, although very reproducible, cannot unambiguously answer the referee's concern, we have complemented the experimental observations with theoretical calculations. In our previous manuscript (Ref. 5) we presented the calculations for the chelation of the FBI molecule with Ba ions surrounded by Xe atoms as well as by Ba perchlorate. Both calculations demonstrated that the most stable configuration was that in which the Ba ions ended in the presenting coordination with one N, phenyl ring and O atoms. Based on this coordination pattern, we assumed that chelation with BaCl₂ behaves in the same way. However, after the referee comment we have decided to compute the chelation of the FBI with BaCl₂ and see what happens if BaCl₂ was located in the periphery of the crown ether instead close to the center.

Thus, we started by locating the BaCl₂ in one of the lateral side by the crown ether and we have calculated the evolution toward the most stable conformations. The calculations were quite demanding and that is the reason for the delay in our response. As it can be seen in Figure 2, the final geometry closely resembles that calculated before. The $\text{N} \cdots \text{Ba}^{2+}$, $\text{Ph} \cdots \text{Ba}^{2+}$ and crown ether $\cdots \text{Ba}^{2+}$ interactions are preserved on passing from naked Ba²⁺ cation to BaCl₂ salt. These geometries show a computed root-mean-square deviation value of RMSD=2.83 Å, excluding the two chloride anions.

Figure 2: Graphical representation of the RMSD associated with fully optimized geometries (B3LYP-D3BJ/6-31G*&LANL2DZ level of theory) of FBI molecule coordinated to a naked Ba²⁺ cation (gray) and to BaCl₂ (green). Hydrogen atoms are not shown.

Taking into account that FBI is on the surface, we have also computed another coordination pattern, forcing the Barium cation to interact with the sensor from the outside, namely through the convex face of the crown ether connected to the fluorophore. In this case, the Ba²⁺ ended interacting only with the crown ether forming a convex complex, as it is shown in Figure 3. In this case, all the O··Ba²⁺ distances are very similar and lie in the 2.83-2.87 Å range. The only N··Ba²⁺ interaction exhibits a slightly larger distance of 3.13 Å. Therefore, this complex has a more restricted coordination pattern since both Ph··Ba²⁺ and N··Ba²⁺ are not available. Consequently, the convex complex is 7.4 kcal/mol (6.9 kcal/mol if Gibbs energies at 298 K are considered) less stable than the previously mentioned concave complex. This means that, thermodynamically, the latter should be ca. 3.0×10^5 more abundant than the latter. However, if the sensor is adsorbed on the surface and the concave face is kinetically less accessible, perhaps the formation of a minor amount of the convex complex cannot be completely ruled out, although even in this rare event, the metastable convex complex should evolve towards the thermodynamically more stable concave one.

Figure 3. Graphical representation of the fully optimized geometry (B3LYP-D3BJ/6-311++G**&LANL2DZ level of theory) of the FBI molecule coordinated to a naked Ba²⁺ along the convex face of the crown ether. Hydrogen atoms are not shown.

To summarize, our simulations indicate that when Ba ions (as ions or coupled to Cl atoms) are placed on the edge of the aza-crown ether, they evolve to establish equivalent bonds with the five oxygen atoms. Therefore, simulations validates the correlation between FBI chelation and O 1s

core level shift, since there are no stable conformations for Ba ions (BaCl₂) at the crown ether periphery (i.e. chemically bonded to one oxygen atom).

ACTIONS TAKEN: In order to clarify this point we have decided to include the following changes:

- Figure 2 has been modified. We have moved the former Figure S1 to Figure 2d and the former Figure 2e to the Supp. Info. In this way, we reinforce the discussion about the chelation using $\theta_{Ba} > 1$.
- Page 10, line 431: We have modified the text to include the new calculations regarding the chelation of FBI with BaCl₂. Here we refer to the passive role of Cl atoms on the final molecular conformation upon Ba chelation.
- Page 11 from lines 473 to line 548: We have reorganized the whole paragraph according to the changes of Figure 2d. As well as makes the discussion about the Ba dose more clear, we have also included a discussion about the absence of 100% chelation, and explaining why we interpret the O 1s core level shift as fingerprint of the chelation. For that purpose, we have included the simulation of the most stable molecular conformation upon chelation in Figure S2, together with calculation details
- We have modified the Figure S3. On the right panel we have included the fitting for the O 1s for the subsequent evaporations on 0.9ML of FBI with BaCl₂. As an inset we have included the Au 4p-O1s core level region.

2. *The authors now provide Ba 3d_{5/2} spectra and conclude that the Ba ion remains in a 2+ oxidation state upon chelation. The spectra only represent a Ba coverage up to $\Theta_{BaBa} = 0.42$. Thus, potential modifications in the spectral signature at saturation (as for example demonstrated in ref. 38 for Na) remain elusive. Additionally, the Ba 3d_{5/2} binding energy seems to be rather insensitive to the oxidation state, with similar binding energies reported for metallic, elemental Ba (<https://srdata.nist.gov/xps/ElmSpectralSrch.aspx?selEnergy=PE>). Thus, the additional data do not provide a proof of selective chelation and the identification of a 2+ oxidation state seems not fully reliable.*

The referee mentions that the absence of change in the Ba core level position does not provide extra information for the chelation process because Ba 3d B.E. is very insensitive to oxidation state. First, we would like to comment that, while it is true that there are different references in literature where metallic and ionic Ba 3d_{5/2} core levels appears at different B.E., databases reported a difference of 1eV between metallic Ba atoms and Ba ions in BaF₂ (the most similar compound to BaCl₂). In the following table, extracted from <https://xps-database.com/barium-spectra-baf2/> we have remarked the average position of Ba 3d_{5/2} for Ba⁰ and Ba²⁺ based on TXL and NIST databases. It is important to remark that these positions have been referred to Cu (2p_{3/2}) BE = 932.62 eV and Au (3d_{7/2}) BE = 83.98 eV. BE (eV) Uncertainty Range: +/- 0.2 eV

Element	Atomic #	Compound	As-Measured by TXL or NIST Average BE	Largest BE	Hydrocarbon C (1s) BE	Source
Ba	56	YBaCuOx	778.3 eV		285.0 eV	The XPS Library
Ba	56	Ba element (N*4)	779.3 eV	780.7 eV	284.8 eV	Avg BE – NIST
Ba	56	Ba-O (N*4)	779.4 eV	779.9 eV	284.8 eV	Avg BE – NIST
Ba	56	Ba-OAc	779.5 eV		285.0 eV	The XPS Library
Ba	56	Ba-CO ₃	779.8 eV		285.0 eV	The XPS Library
Ba	56	Ba-CO ₃ (N*2)	779.8 eV	779.9 eV	284.8 eV	Avg BE – NIST

Ba	56	Ba-F2 (N*2)	779.8 eV	781.7 eV	284.8 eV	Avg BE – NIST
Ba	56	Ba-S (N*1)	779.8 eV		284.8 eV	Avg BE – NIST
Ba	56	Ba-F2	780.4 eV		285.0 eV	The XPS Library
Ba	56	Ba-O	780.2 eV		285.0 eV	The XPS Library
Ba	56	Ba-SO4	780.7 eV		285.0 eV	The XPS Library
Ba	56	Ba-O2 (N*1)	780.8 eV		284.8 eV	Avg BE – NIST
Ba	56	Ba-(ClO4)2 (N*1)	780.9 eV		284.8 eV	Avg BE – NIST
Ba	56	Ba-H2 (N*1)	782.9 eV		284.8 eV	Avg BE – NIST
Ba	56	Ba-(OH)2			285.0 eV	The XPS Library

In our work, we are comparing the energy position with our own measurements of BaCl₂ deposited on Au, where we are sure that 2+ oxidation state is preserved. In our case, if chelation works, we expected no change in the Ba core level position, because it should remain in the 2+ oxidation state. And this is the case, we observe no change in the core level position for the subsequent sublimations on FBI molecules, while the ratio Cl:Ba changes.

The referee mentions Reference 38, as example where, after saturation of crown ether, there is a monotonous shift of the Na core level, due to doping. In Ref. 38 crown ether molecules are co-deposited with porphyrin molecules. Both molecules have affinity to capture ions, and authors discern the preferential chelation of crown ether group versus porphyrin rings by the different shift in the Na core level position. During crown ether chelation the Na core level remains unchanged (up to $\theta=0.8$) and then shift when Na atoms interact with the porphyrins (they talked about doping instead of coordination for porphyrins). However, in our case, we only have one kind of molecule and, if we increase the coverage above saturation, we will just have BaCl₂ on the surface.

ACTION TAKEN: We have included the following sentence on page 10, line 446: “In any case, the Ba 3d core level does not show any significant shift when comparing the sublimation of BaCl₂ on bare or functionalized gold surface (Figure S3 in the Supp. Info), suggesting that there is no change in the Barium chemical oxidation state. [43]”

3. *The addition of the electron density contours is instructive. Nonetheless, the authors now indicate that an "unambiguous association of chelation and molecular rearrangement" can not be done by STM, but only by STS. Thus, additional STS data providing some statistics on several molecules and a refined analysis (e.g., addressing the uncertainty in the determination of the HOMO onset for FBI-Na+) would be important to substantiate the conclusions about chelation and molecular rearrangement. It also does not seem immediately clear why spectra recorded on the fluorophore region, which is claimed to be planar and interacting with the Au support in all three cases, should be representative for the structural rearrangements that mainly occur on the crown ether moiety (where also the apparent height is measured). Even though Figure 4 was corrected, the statement "... were measured well inside the molecular gap" still seems not really appropriate. Fig. 4e) (green spectrum) suggests that the tail of the LUMO contributes at a bias voltage of 1.4 V (c).*

Following the referee’s requirement, we have included extra STS spectra of chelated Na and Ba chelated FBI molecules. After the first request of the reviewer, we included in the Supporting Information a figure with extra STS spectra for non-chelated FBI molecules. At that time, the spectra were measured for molecules at the island edges. However, for chelated species this was not possible (tip was not stable when we tried to do it) so, the extra STS spectra that we have now included in the Figure S5 correspond all of them to individual molecules.

All the spectra were measured at the fluorophore region because performing spectroscopy and/or well resolved images at the crown ether regions was not possible because of their very high flexibility. The frontier orbitals calculated from DFT are mainly located at the fluorophore region for both free and chelated molecules (calculations were done for our previous work, Ref. 5). Thus, although the molecular orbital could be slightly modified by hybridization with the substrate (which could affect the HOMO-LUMO positions), the changes we measured are in agreement with the variation in the frontier orbitals that theory predicted comparing before and after chelation (chelated species has a higher gap compared to unchelated one).

Regarding the last comment referring to the fact that Image 4c was measured at 1.4V of bias voltage, we should comment that measuring at lower voltage was complicated because the molecule was not stable and we could not obtain reproducible and reliable images.

ACTIONS TAKEN:

We have modified the Figure S5 to include the new spectra.

- Page 14, line 633: We have rephrased the sentence "... were measured well inside the molecular gap" by "measured using bias voltages inside the molecular gap to reduce as much as possible the influence of the electronic structure"
- Page 15, line 663 We have rephrased the sentence "Even with this evidence, unambiguous association apparent molecular structural modification and chelation can only be done by comparing the spectroscopy scanning tunneling spectroscopy (STS)" by "To establish a more quantitative relationship between apparent molecular structural modification and chelation, scanning tunneling spectroscopy (STS) measurements were performed."

4. *The description of the comparison of STS and spectroscopy in solution was improved. Nonetheless, some of the statements still seem confusing and not fully supported by the data. Focusing on the trend of the gap, table 1 shows considerable blue shifts for Ba-chelated FBI (49.5 nm), but also for Na-chelated FBI (16.5 nm). The optical spectra mainly show a blue shift for Ba-chelated FBI (Emission Ba: 61 nm, Na: 0 nm). The conclusion reads "Nonetheless, the measured variations in the molecular HOMO - LUMO gap are perfectly consistent with the observed emission shift for Ba 2+. Moreover, they are also consistent with the absence of such shift, observed and predicted for other ions, such as Na+." While one might argue about the term "perfectly consistent" when discussing a trend, the second sentence seems to contradict the blue shift observed in the HOMO - LUMO gap for Na. And it also does not seem in line with the statement on page 15 "Thus, the measured HOMO - LUMO gap values show a blue shift tendency for the chelated molecules, mainly for Ba2+-chelated FBI, which follows the trends observed in fluorescence spectroscopy for the same molecules in solution". As the authors emphasize the important "bicolour property of the sensor" and as the discrimination of ions is crucial for the detection scheme, this issue is relevant.*

Since the explanation of this section still induces to error, we have rephrased it to make it clearer. It is true that it is difficult to correlate the shift in the HOMO-LUMO gap and the shift in the fluorescence spectroscopy, moreover, having into account that STS is measured for molecules on the surface and fluorescence for molecules in solution. In the revised text version, we now explain in more details which are the de-excitation transitions for fluorescence spectroscopy and we relate the STS observation with the DFT calculations of the frontier molecular orbitals.

And even when we agree that the bicolor property is the differential aspect of the sensor, the discrimination of possible undesirable chelation with other ions than Barium is not critical for the final detector. The NEXT experiment is being designed to reduce as much as possible these eventual events. First, as we explained in our previous answer letter, by controlling by RGA the

nature of the residual background of the experimental chamber. Second, and more importantly, by designing a trigger protocol, activated by the Xe scintillation, to activate the sensor, preventing undesirable chelation.

ACTION TAKEN: Page 15 from line 669 to 734: We have modified the discussion about the STS results and how they are related with fluorescence spectroscopy.

Minor comments:

- It might be appropriate to mention the Ba²⁺ coverage used for the STM imaging in Fig. 4.- The referee does not recognize the benefits of using a colored background in Fig. 2d/e and Fig. 4e. It makes it hard to read the labels (Fig. 2) and the additional colors (Fig. 4), not matching the colors of the spectra, are rather confusing.- Certain wordings seem a bit strange. Line 312: "approximately resolved"; line 490 "core level was also controlled."- The referee suggests to refrain from citing unpublished work in the conclusions.

Additional comment: Reviewers invest quite some efforts in assessing this manuscript, as a service for the authors. Accordingly, authors would be expected to carefully read their manuscript before submission, as a service for the referees. This does not seem to be the case for this manuscript, which includes several careless mistakes. Examples:

Line 299: S1 should read S3

Line 310: "On the right" isn't this "on the left"?

Line 314: "region is" or "regions are"

Fig. 2: inconsistent notation (Θ versus Φ)

Line 442: bracket and punctuation mark missing

Line 451: punctuation mark missing

Line 494: S1 should read S2

Line 626: "the spectroscopy scanning tunneling spectroscopy"

No reference to Fig. S4 in the main text

Caption Fig. S3: "Red squares show..", there are no red squares in the figure.

Caption Fig. S5: Does the bottom panel in b) really represent the electron density associated with the conformer show in a)? The labeling seems wrong.

We appreciate the minor changes and comments that referee suggests. We have taken care of all of them, including the color changes for Figure 4.

REVIEWERS' COMMENTS

Reviewer #3 (Remarks to the Author):

The authors modified the manuscript to address the remaining issues from the previous report. Important aspects, such as the suitability of the O 1s XPS signal as fingerprint for chelation, the interpretation of the STS data, and the comparison to optical spectroscopy were clarified. The modifications include results from additional calculations.

With minor exceptions that cause some confusion ("In fact in any of the experiments, we have performed so far, the 100% chelation was observe" versus "It is important to note that .. we never observed a complete chelation of the FBI."), the response letter is fully comprehensible and the manuscript was further improved.

In the reviewer's opinion, the manuscript can be accepted for publication in Nature Communications.